# SageAttention3: Microscaling FP4 Attention for Inference and An Exploration of 8-bit Training

**Jintao Zhang**[*12], **Jia Wei**[*1], **Haoxu Wang**[1], **Pengle Zhang**[1], **Xiaoming Xu**[1], **Haofeng Huang**[1], **Kai Jiang**[1], **Jianfei Chen**[†1], **Jun Zhu**[†12]

[1]Dept. of Comp. Sci. and Tech., Institute for AI, BNRist Center, THBI Lab,
Tsinghua-Bosch Joint ML Center, Tsinghua University;    [2]Shengshu Tech., Beijing, China.
{zhang-jt24@mails., jianfeic@, dcszj@}tsinghua.edu.cn

## Abstract

The efficiency of attention is important due to its quadratic time complexity. We enhance the efficiency of attention through two key contributions: First, we leverage the new FP4 Tensor Cores in Blackwell GPUs to accelerate attention computation. Our implementation achieves **1038** TOPS on RTX5090, which is a **5×** speedup over the fastest FlashAttention on RTX5090. Experiments show that our FP4 attention can accelerate inference of various models in a plug-and-play way. Second, we pioneer low-bit attention to training tasks. Existing low-bit attention works like FlashAttention3 and SageAttention focus only on inference. However, the efficiency of training large models is also important. To explore whether low-bit attention can be effectively applied to training tasks, we design an accurate and efficient 8-bit attention for both forward and backward propagation. Experiments indicate that 8-bit attention achieves lossless performance in fine-tuning tasks but exhibits slower convergence in pretraining tasks. The code is available at https://github.com/thu-ml/SageAttention.

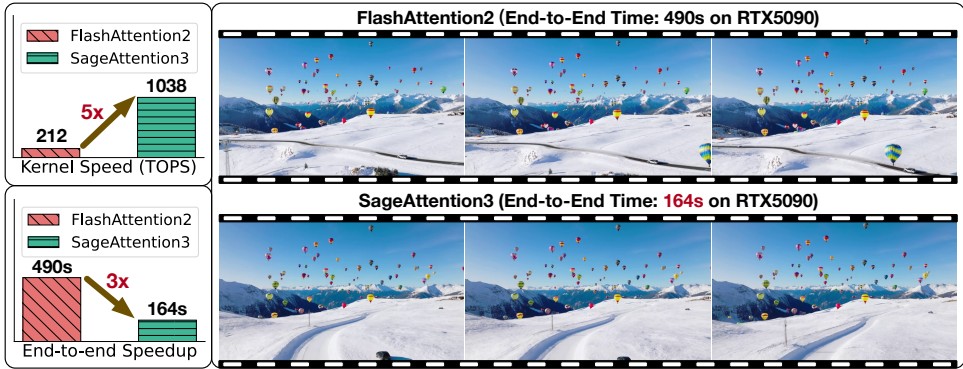

Figure 1: The upper left figure shows the kernel speedup on RTX5090. The other two figures show the end-to-end inference speedup of generating a video using HunyuanVideo on RTX5090. Note that FlashAttention3 can only run on Hopper GPUs, so FlashAttention2 is already the fastest on RTX5090.

## 1   Introduction

**Motivation**. The efficiency of attention is critical for generation models, especially given their quadratic time complexity with longer sequences [1, 2]. Quantization offers an effective way to

---

[*] co-first authors, [†] Corresponding authors.

39th Conference on Neural Information Processing Systems (NeurIPS 2025).

accelerate inference by utilizing low-bit Tensor Cores in GPUs [3, 4, 5, 6]. The new FP4 Tensor Cores in Blackwell GPUs deliver significantly faster performance compared to FP16 [7]. We want to propose a novel FP4 attention implementation that provides plug-and-play compatibility for inference acceleration. Beyond inference, training efficiency is equally important. However, no prior work has explored low-bit attention for training large models. To address this gap, we design a trainable 8-bit attention to explore its feasibility in training tasks.

To the best of our knowledge, we are the first work that designs FP4 attention for inference and the first work to explore the feasibility of low-bit attention for training large models.

**Challenges.** There are two primary obstacles for FP4 attention and one key difficulty for 8-bit trainable attention. First, **(C1)** FP4 quantization suffers from severe value limitations (only 15 representable values), making both per-tensor and per-token quantization approaches inadequate for preserving model accuracy. Second, **(C2)** The attention map $P$ consists primarily of small values in the range $[0, 1]$. When directly quantized to FP4, these values force the scaling factors into an extremely narrow dynamic range. However, hardware requires the quantization factors to be in FP8 data type. This leads to significant accuracy loss when presenting these scale factors in FP8. Third, **(C3)** When employing 8-bit attention during training, we find that the attention map gradients are particularly vulnerable to quantization errors, resulting in accumulated errors in the input gradients.

**Our Method.** To address **(C1)**, we propose to use FP4 microscaling quantization for the two matrix multiplications in attention, i.e., $QK^\top$ and $PV$. By constraining the quantization group size to 1x16 (instead of per-tensor or per-channel), our method effectively contains outlier effects within each block while improving FP4 quantization accuracy. To overcome **(C2)**, we propose a two-level quantization method for $P$ to fully utilize the presentative range of the FP8 scaling factor, enhancing the quantization accuracy of $P$. Specifically, this approach first normalizes each token's range to $[0, 448 \times 6]$ through per-token quantization, then applies FP4 microscaling quantization for enhanced precision. To address **(C3)**, we identify the most accuracy-sensitive matrix multiplication among the five in backpropagation and maintain its accuracy in FP16.

**Result.** Our FP4 attention, named SageAttention3, could achieve **1038** TOPS on RTX5090, which is a **5×** speedup than FlashAttention. Furthermore, we demonstrate that 8-bit trainable attention, named SageBwd, could achieve lossless performance when fine-tuning base models for instruction-following tasks, but is not suitable for pretraining tasks.

**Contribution.** Our work makes the following key contributions:

**(1)** We design the first FP4 attention to accelerate inference, achieving **1000+** TOPS on RTX5090.

**(2)** We propose the first trainable low-bit attention, enabling accelerated training with lossless fine-tuning performance, while revealing key insights for low-bit attention in training.

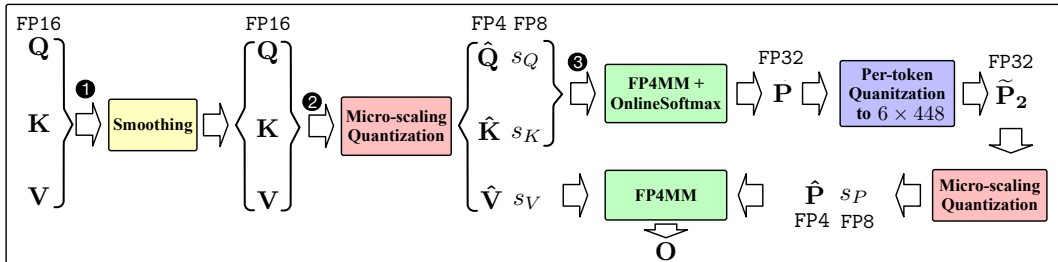

Figure 2: Workflow of microscaling FP4 attention.

## 2 Preliminary

**FlashAttention.** The attention computation contains two matrix multiplications and one softmax calculation: $S = QK^\top, P = \texttt{Softmax}(S), O = PV$. The $Q, K, V$ are in the shape of $N \times D$, where $N$ means the sequence length and $D$ means the dimension of an attention head. $P, S$ are in the shape of $N \times N$. FlashAttention divides $Q$ to blocks $\{Q_i\}$ in the shape of $B_q \times D$, and divides $K, V$ to $\{K_i\}, \{V_i\}$ in the shape of $B_{kv} \times D$. Then it uses online softmax to avoid the large memory IO for $S$ and $P$: $S_{ij} = Q_i K_j^\top, P_{ij} = \texttt{OnlineSoftmax}(S_{ij}), O_{ij} = P_{ij}V_j$.

**Notation.** For simplicity, we omit subscripts and use $\mathbf{Q}, \mathbf{K}, \mathbf{V}, \mathbf{S}, \mathbf{P}, \mathbf{O}$ to denote the matrix blocks in FlashAttention, while retaining full subscript notation in Algorithm 1, 2, and 3.

**Quantization.** Quantization is used to accelerate Matmul by converting two matrices from high-bit to low-bit with scale factors. Take `INT8` quantization for Matmul $AB$ as an example, where $A$ and $B$ are in `FP16` data type. It can be formulated: $s_A = \max(|A|)/127$, $\hat{A} = \lceil A/s_A \rceil$, $s_B = \max(|B|)/127$, $\hat{B} = \lceil B/s_B \rceil$, where $\hat{A}, \hat{B}$ are in `INT8` and the others are in `FP32`. Then, $AB \approx \hat{A}\hat{B} \times s_A \times s_B$, which can be accelerated by the `INT8` Tensor Core. The granularity of quantization is determined by the dimensions reduced by the max operation. For example, in *per-token quantization*, the max is computed along each row of a matrix. In *per-block quantization*, the max is computed on a block of a matrix, which in our paper means a FlashAttention block.

## 3 FP4 Attention for Inference Acceleration

This section presents our microscaling `FP4` attention through three key components: (1) the fundamental workflow for applying microscaling `FP4` quantization to attention in Section 3.1, (2) the two-level quantization approach for the attention map in Section 3.2, and (3) critical hardware implementation optimization in Section 3.3.

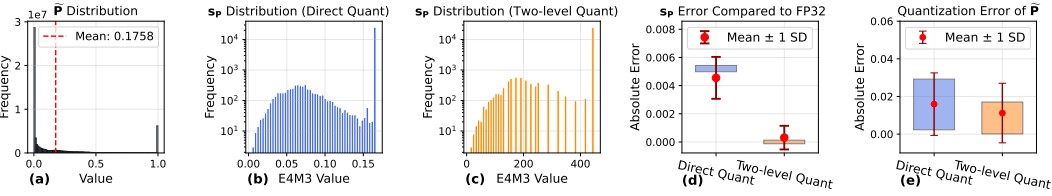

Figure 3: Analysis of the benefit of two-level quantization. (a) shows the distribution of $\widetilde{\mathbf{P}}$. (b) and (c) show the distribution of $\mathbf{s_P}$ using direct quantization and two-level quantization. (d) and (e) show the error of $\mathbf{s_P}$ and $\widetilde{\mathbf{P}}$ using direct quantization and two-level quantization.

### 3.1 Microscaling FP4 Attention

**FP4 microscaling quantization**. Given a matrix $X \in \mathbb{R}^{N \times d}$, we quantize it to $\hat{X}$ in `FP4` data type with a scale factor matrix $s_X$ in `FP8` data type. Specifically, $X$ is partitioned into $X_{ij} \in \mathbb{R}^{1 \times n}$ blocks, where each $1 \times n$ block corresponds to one scale factor $s_{ij}$. The `FP4` microscaling quantization ($[\hat{X}, s_X = \phi(X)]$) and dequantization ($X' = \phi^{-1}(\hat{X}, s_X)$) can be formulated as follows.

$$\text{Quantization } \phi: s_{ij} = \max(|X|)/6, \quad \hat{X}_{ij} = \lceil X_{ij}/s_{ij} \rceil \tag{1}$$

$$\text{Dequantization } \phi^{-1}: X'_{ij} = s_{ij} \times \hat{X}_{ij} \tag{2}$$

Where the $\lceil \cdot \rceil$ means `FP4` rounding.

**FP4 microscaling quantization Matmul**. Consider a matrix multiplication $AB$, where $A$ and $B$ are in `FP16` precision. The speed of the Matmul is about 200 `TOPS` on `RTX5090`. In contrast, the speed of the `FP4` microscaling Matmul is about 1600 `TOPS`, which is an 8x speedup. The `FP4` microscaling Matmul instruction (`FP4MM`) takes four inputs, i.e., $\hat{A}, s_A, \hat{B}, s_B$, and the output $C$ equals to the Matmul result between $\phi^{-1}(\hat{A}, s_A)$ and $\phi^{-1}(\hat{B}, s_B)$:

$$C = \texttt{FP4MM}(\hat{A}, s_A, \hat{B}, s_B) \tag{3}$$

**Attention computation.** We accelerate attention computation by applying `FP4` microscaling quantization to both matrix multiplications: $\mathbf{Q}\mathbf{K}^\top$ and $\mathbf{P}\mathbf{V}$.

$$\hat{\mathbf{Q}}, \mathbf{s_Q} = \phi(\mathbf{Q}), \quad \hat{\mathbf{K}}, \mathbf{s_K} = \phi(\mathbf{K}^\top), \quad \mathbf{S} = \texttt{FP4MM}(\hat{\mathbf{Q}}, \mathbf{s_Q}, \hat{\mathbf{K}}, \mathbf{s_K})$$

$$\widetilde{\mathbf{P}} = \texttt{OnlineSoftmax}(\mathbf{S})$$

$$\hat{\mathbf{P}}, \mathbf{s_P} = \phi(\widetilde{\mathbf{P}}), \quad \hat{\mathbf{V}}, \mathbf{s_V} = \phi(\mathbf{V}), \quad \mathbf{O} = \texttt{FP4MM}(\hat{\mathbf{P}}, \mathbf{s_P}, \hat{\mathbf{V}}, \mathbf{s_V}) \tag{4}$$

It is important to note that our hardware implementation builds on FlashAttention, where the matrices $\mathbf{Q}$, $\mathbf{K}$, $\widetilde{\mathbf{P}}$, and $\mathbf{V}$ in our formulation correspond to FlashAttention's tiled $Q$, $K$, $\widetilde{P}$, and $V$ blocks as described in Section 2. Additionally, to enhance the attention accuracy, we adopt the smoothing $Q$ and $K$ in SageAttention2 [4]. The complete algorithm is presented in Algorithm 1.

**Data type determination.** There are two choices for the FP4 data type [8]. The first one is the NVFP4, which is in E2M1 data type and its quantization block size is $1 \times 16$ and its scale factor is in E4M3 data type. The second one is the MXFP4, which is also in E2M1 data type. However, its quantization block size is $1 \times 32$ and its scale factor is in E8M0 data type. We choose NVFP4 because the accuracy of NVFP4 is much higher than that of MXFP4 in attention quantization. Empirical results: Table 1(a) shows the accuracy of MXFP4 and NVFP4 using real $\mathbf{Q, K, V}$ across all layers of CogVideoX. Results indicate that the accuracy of NVFP4 outperforms that of MXFP4.

---

**Algorithm 1:** Implementation of the microscaling FP4 attention.

1: **Input:** Matrices $Q(\texttt{FP16})$, $K(\texttt{FP16})$, $V(\texttt{FP16}) \in \mathbb{R}^{N \times d}$, block size $B_q$, $B_{kv}$.
2: **Preprocessing:** $K = K - \text{mean}(K)$ // Smoothing K of SageAttention.
3: Divide $Q$ to $T_m = N/B_q$ blocks $\{\mathbf{Q}_i\}$; divide $K$, and $V$ to $T_n = N/B_{kv}$ blocks $\{\mathbf{K}_i\}$, $\{\mathbf{V}_i\}$ ;
4: **for** $i = 1$ **to** $T_m$ **do**
5: $\quad \bar{q}_i = \text{mean}(\mathbf{Q}_i)$, $(\mathbf{s_Q}, \hat{\mathbf{Q}}_i) = \phi(\mathbf{Q}_i - \bar{q}_i)$ ; // Smoothing Q of SageAttention2.
6: $\quad$ **for** $j$ in $[1, T_n]$ **do**
7: $\quad\quad (\mathbf{s_K}, \hat{\mathbf{K}}_j) = \phi(\mathbf{K}_j^\top)$ , $(\mathbf{s_V}, \hat{\mathbf{V}}_j) = \phi(\mathbf{V}_j)$ ;
8: $\quad\quad \mathbf{S}_{ij} = \texttt{FP4MM}(\hat{\mathbf{Q}}_i, \mathbf{s_Q}, \hat{\mathbf{K}}_j, \mathbf{s_K}) + \texttt{GEMV}(\bar{q}_i, \mathbf{K}_j^\top)$ ; // Smoothing Q.
9: $\quad\quad m_{ij} = \max(m_{i,j-1}, \text{rowmax}(\mathbf{S}_{ij}))$, $\widetilde{\mathbf{P}}_{ij} = \exp(\mathbf{S}_{ij} - m_{ij})$,
$\quad\quad l_{ij} = e^{m_{i,j-1} - m_{ij}} l_{i,j-1} + \text{rowsum}(\widetilde{\mathbf{P}}_{ij})$ ;
10: $\quad\quad \mathbf{s_{P_1}} = \text{rowmax}(\widetilde{\mathbf{P}}_{ij})/(448 \times 6)$, $\widetilde{\mathbf{P}}_{ij} = \widetilde{\mathbf{P}}_{ij}/\mathbf{s_{P_1}}$, $\mathbf{s_{P_2}}, \hat{\mathbf{P}}_{ij} = \phi(\widetilde{\mathbf{P}}_{ij})$; // two-level quantization
11: $\quad\quad \mathbf{O}_{ij} = \text{diag}(e^{m_{i,j-1} - m_{ij}})\mathbf{O}_{i,j-1} + \texttt{FP4MM}(\hat{\mathbf{P}}_{ij}, \mathbf{s_{P_2}}, \hat{\mathbf{V}}_j, \mathbf{s_V}) \times \mathbf{s_{P_1}}$
12: $\quad$ **end for**
13: $\quad \mathbf{O}_i = \text{diag}(l_{i,T_n})^{-1}\mathbf{O}_{i,T_n}$ ;
14: **end for**
15: **return** $O = \{\mathbf{O}_i\}$

---

## 3.2 Two-level Scaling for $\widetilde{\mathbf{P}}$

Applying microscaling FP4 quantization for $\widetilde{\mathbf{P}}$ presents a challenge to attention accuracy. For example, Fig. 12( c) shows direct quantization severely degrades output quality, producing results substantially different from full-precision outputs. Our analysis reveals that the issue occurs because microscaling NVFP4 quantization requires the scale factor to be represented in E4M3 FP8 format [9], rather than the FP32 data type typically used for scale factors. This causes accuracy loss when the scale factor is directly converted to E4M3 format. To better understand this accuracy loss, we analyze the data distribution of $\widetilde{\mathbf{P}}$ and its scale factors in Fig. 3. Since $\widetilde{\mathbf{P}}$ is computed using online softmax [10], the values in each microscaling block $\widetilde{\mathbf{P}}_{ij}$ fall $[0, 1]$. Consequently, the scale factor (scale factor $= \max(\widetilde{\mathbf{P}}_{ij})/6$) ranges between 0 and 0.167. This narrow range leads to inefficient usage of E4M3's representable range, increasing accuracy loss. To reduce accuracy loss by fully utilizing E4M3's range, we propose a two-level quantization method for the $\widetilde{\mathbf{P}}$ matrix. Specifically, we first quantize each row of $\widetilde{\mathbf{P}}$ to $[0, 448 \times 6]$. Then we apply the standard FP4 quantization $\phi$ for the quantized $\widetilde{\mathbf{P}}$. The two-level quantization can be formulated as follows:

$$\mathbf{s_{P_1}} = \text{rowmax}(\widetilde{\mathbf{P}})/(448 \times 6), \quad \widetilde{\mathbf{P}}_2 = \widetilde{\mathbf{P}}/\mathbf{s_{P_1}}, \quad \mathbf{s_{P_2}}, \hat{\mathbf{P}}_2 = \phi(\widetilde{\mathbf{P}}_2)$$
$$(\widetilde{\mathbf{P}} \approx \hat{\mathbf{P}}_2 \times \mathbf{s_{P_2}} \times \mathbf{s_{P_1}}), \quad \mathbf{O} = \texttt{FP4MM}(\hat{\mathbf{P}}_2, \mathbf{s_{P_2}}, \hat{\mathbf{V}}, \mathbf{s_V}) \times \mathbf{s_{P_1}} \quad (5)$$

Where $\widetilde{\mathbf{P}}$, $\widetilde{\mathbf{P}}_2$, and $\mathbf{s_{P_1}}$ are in FP32 data type. $\mathbf{s_{P_2}}$ and $\mathbf{s_V}$ are in FP8 data type. $\hat{\mathbf{P}}_2$ and $\hat{\mathbf{V}}$ are in FP4 data type.

Empirical results: As shown in Fig. 3, our two-level quantization maximizes the E4M3 range utilization for $\mathbf{s_P}$, thereby reducing both the numerical representation error of $\mathbf{s_P}$ and the quantization error of $\widetilde{\mathbf{P}}$. A more formal theoretical analysis is provided in Appendix A.5. Table 1(b) shows the accuracy of

two-level quantization against naive direct quantization, using real Q, K, V from layers of CogVideoX. Results indicate that two-level quantization boosts the accuracy.

### 3.3 Implementation and Optimization on Hardware

**Permutation for K.** Unlike `FP16`, the `FP32` accumulator's memory layout in FP4 MatMul [11] differs from its operand A's register layout (shown in Fig. 20 and 19). Performing thread shuffles to match operand A's layout would degrade kernel performance. Our solution transforms the accumulator layout (Fig. 21) by permuting the P tile's columns. To maintain correct MatMul, we correspondingly rearrange K's columns, which can be fused with the quantization kernel.

**Reuse shuffle.** The in-kernel micro-scaling quantization of $\widetilde{\mathbf{P}}$ requires finding the max value of 16 consecutive row elements. However, as shown in Fig. 21, these 16 elements are distributed across four threads, necessitating intra-thread max reduction followed by inter-thread shuffling, significantly slowing down the kernel. We optimize this by fusing quantization with online softmax, which also computes row-wise maxima. First, we compute the max over 16 elements in $S$ and reuse it in the subsequent softmax max-reduction. This fusion reduces redundant shuffles and max operations by 50%, yielding about 10% whole kernel speedup.

**Producer warp epilogue.** In conventional warp-specialized kernels, consumer warps typically handle both MatMul and store operations while producers merely load inputs, with ping-pong scheduling between consumers enabling stage overlap [12]. However, register constraints make this approach infeasible for our `FP4` attention kernel. Instead, we implement ping-pong scheduling between producer warps: while one producer loads inputs for the next MatMul operation, another concurrently stores outputs to global memory, with consumer warps solely responsible for transferring MatMul results from registers to shared memory. This novel design overlaps MatMul and global memory stores within register constraints, boosting throughput.

## 4 INT8 Attention for Training

Low-bit quantization attention works, such as FlashAttention3 and SageAttention, are only for inference. In this section, we propose an `INT8` attention for training, named `SageBwd`, which quantizes six of seven matrix multiplications in attention to `INT8`, achieving no performance degradation in fine-tuning tasks. Besides, we implement both `INT8 SageBwd` and `FP8 SageBwd` and conduct comparison experiments, proving `INT8 SageBwd` is superior to `FP8 SageBwd` in Section 5.4.

---

**Algorithm 2:** Forward pass of the `8-bit` attention.

---

1: **Input:** FP16 matrices $Q, K, V \in \mathbb{R}^{N \times d}$, and block size $B_q, B_{kv}$.
2: $K_m = \text{mean}(K); \quad K \leftarrow K - K_m$ ; // Smooth-k technique.
3: Divide $Q$ to $T_m = N/B_q$ blocks $\{\mathbf{Q}_i\}$; divide $K$, and $V$ to $T_n = N/B_{kv}$ blocks $\{\mathbf{K}_i\}, \{\mathbf{V}_i\}$ ;
4: **Quantization:** $\{\mathbf{s_Q}, \hat{\mathbf{Q}}_i\} = \{\psi(\mathbf{Q}_i)\}, \quad \{\mathbf{s_K}, \hat{\mathbf{K}}_i\} = \{\psi(\mathbf{K}_i^\top)\}, \quad \{\mathbf{s_V}, \hat{\mathbf{V}}_i\} = \{\psi(\mathbf{V}_i)\}$ ; // Per-block.
5: **for** $i = 1$ **to** $T_m$ **do**
6:     $\mathbf{O}_i \in \mathbb{R}^{B_q \times D} = (0), \quad \mathbf{L}_i \in \mathbb{R}^{B_q} = (0), \quad m_i \in \mathbb{R}^{B_{kv}} = (0)$ ;
7:     **for** $j$ in $[1, T_n]$ **do**
8:         $\mathbf{S}_{ij} = \text{MM}(\hat{\mathbf{Q}}_i, \hat{\mathbf{K}}_j) \times \mathbf{s_Q} \times \mathbf{s_K}$ ;
9:         $m_{ij} = \max(m_{i,j-1}, \text{rowmax}(\mathbf{S}_{ij})), \widetilde{\mathbf{P}}_{ij} = \exp(\mathbf{S}_{ij} - m_{ij}),$
        $l_{ij} = e^{m_{i,j-1} - m_{ij}} l_{i,j-1} + \text{rowsum}(\widetilde{\mathbf{P}}_{ij});$
10:         $\mathbf{s_P} = \exp(\text{rowmax}(\mathbf{S}_{ij}) - m_{ij})/127, \quad \hat{\mathbf{P}}_{ij} = \widetilde{\mathbf{P}}_{ij}/\mathbf{s_P}$ ; // Per-token quantization.
11:         $\mathbf{O}_{ij} = \text{diag}(e^{m_{i,j-1} - m_{ij}})\mathbf{O}_{i,j-1} + \text{MM}(\hat{\mathbf{P}}_{ij}, \hat{\mathbf{V}}_j) \times \mathbf{s_P} \times \mathbf{s_V}$
12:     **end for**
13:     $\mathbf{O}_i = \text{diag}(l_{i,T_n})^{-1}\mathbf{O}_{i,T_n}$ ;
14:     $\mathbf{L}_i = m_{i,T_n} + \log(l_{i,T_n})$ ;
15: **end for**
16: **return** $O = \{\mathbf{O}_i\}, L = \{\mathbf{L}_i\}$ ;

---

## 4.1 Forward

There are two matrix multiplications in the forward pass of attention:

$$\mathbf{S} = \mathbf{Q}\mathbf{K}^\top, \ \ \mathbf{O} = \mathbf{P}\mathbf{V} \tag{6}$$

**Per-token quantization for P.** Following SageAttention [3], we apply smoothing K and per-block `INT8` quantization for the $\mathbf{Q}\mathbf{K}^\top$. However, for the $\widetilde{\mathbf{P}}\mathbf{V}$, a static per-block `INT8` quantization with a static scale factor of $1/127$ for $\widetilde{\mathbf{P}}$ is inaccurate [3]. Fortunately, we find applying per-token `INT8` quantization for $\widetilde{\mathbf{P}}\mathbf{V}$ and per-block `INT8` quantization for $\mathbf{V}$ can enhance the attention accuracy. Furthermore, we eliminate the need for explicit max operations on $\mathbf{P}$ by reusing both global and local maximum values from the online softmax computation (Line 9 in Algorithm 2). The algorithm for the forward is shown in Algorithm 2.

Given our extensive use of `INT8` per-block quantization in trainable attention, we formalize the process as follows. For each FlashAttention block $\mathbf{X}$, the quantization process $\mathbf{s_X}, \hat{\mathbf{X}} = \psi(\mathbf{X})$ can be formulated as:

$$\mathbf{s_X} = \max(|\mathbf{X}|)/127, \ \ \hat{\mathbf{X}} = \mathbf{X}/\mathbf{s_X} \tag{7}$$

---

**Algorithm 3:** Backward pass of the `8-bit` attention.

1: **Input:** $\{\mathbf{s_Q}, \hat{\mathbf{Q}}_i\}, \{\mathbf{s_K}, \hat{\mathbf{K}}_i\}, \{\mathbf{s_V}, \hat{\mathbf{V}}_i\}, K_m, O, \{\mathbf{L}_i\}$ from forward, $dO \in \mathbb{R}^{N \times d}$, block size $B_q, B_{kv}$ ;
2: $D = \text{rowsum}(dO \circ O)$, divide $D$ to $T_m = N/B_q$ blocks $\{\mathbf{D}_i\}$;
3: **for** $j = 1$ to $T_n$ **do**
4:     **for** $i$ in $[1, T_m]$ **do**
5:         $\mathbf{S}_{ij} = \text{MM}(\hat{\mathbf{Q}}_i, \hat{\mathbf{K}}_j) \times \mathbf{s_Q} \times \mathbf{s_K}$ ; $\ \ \ \mathbf{P}_{ij} = \exp(\mathbf{S}_{ij} - \mathbf{L}_i)$ ;
6:         $\mathbf{s_P}, \hat{\mathbf{P}}_{ij} = \psi(\mathbf{P}_{ij})$, $\ \ \mathbf{s_{dO}}, \hat{\mathbf{dO}}_i = \psi(\mathbf{dO}_i)$ ; // `INT8` per-block quantization.
7:         $\mathbf{dV}_j \leftarrow \mathbf{dV}_j + \text{MM}(\hat{\mathbf{P}}_{ij}^\top, \hat{\mathbf{dO}}_i) \times \mathbf{s_P} \times \mathbf{s_{dO}}$ ;
8:         $\mathbf{dP}_{ij} = \text{MM}(\mathbf{dO}, \mathbf{V}_j^\top)$ ; // Keep in FP16.
9:         $\mathbf{dS}_{ij} = \mathbf{P}_{ij} \circ (\mathbf{dP}_{ij} - \mathbf{D}_i)$ ; $\ \ \ \mathbf{s_{dS}}, \hat{\mathbf{dS}}_{ij} = \psi(\mathbf{dS}_{ij})$ ; // `INT8` per-block quantization.
10:         $\mathbf{dQ}_i \leftarrow \mathbf{dQ}_i + \text{MM}(\hat{\mathbf{dS}}_{ij}, \hat{\mathbf{K}}_j) \times \mathbf{s_{dS}} \times \mathbf{s_K} + \text{rowsum}(\mathbf{dS}_{ij})K_m$ ; // Backward for smooth-k.
11:         $\mathbf{dK}_j \leftarrow \mathbf{dK}_j + \text{MM}(\hat{\mathbf{dS}}_{ij}^\top, \hat{\mathbf{Q}}_i) \times \mathbf{s_{dS}} \times \mathbf{s_Q}$ ;
12:     **end for**
13: **end for**
14: **return** $dQ, dK, dV$ ;

---

## 4.2 Backward

There are five matrix multiplications in the backward pass of attention:

$$\mathbf{S} = \mathbf{Q}\mathbf{K}^\top, \ \ \mathbf{dV} = \widetilde{\mathbf{P}}^\top \mathbf{dO}, \ \ \mathbf{dP} = \mathbf{dO}\mathbf{V}^\top, \ \ \mathbf{dQ} = \mathbf{dS}\mathbf{K}, \ \ \mathbf{dK} = \mathbf{dS}^\top \mathbf{Q} \tag{8}$$

We observe that whether applying quantizing to $\mathbf{dO}\mathbf{V}^\top$ has a significant impact on the accuracy of the gradient of $Q, K$. This is because the accuracy of $\mathbf{dO}\mathbf{V}^\top$ directly determines the accuracy of $\mathbf{dP}$ and $\mathbf{dS}$ (see computational dependencies in Algorithm 3). The accuracy loss in $\mathbf{dS}$ will continuously accumulate errors into $\mathbf{dQ}$ and $\mathbf{dK}$ during the recurrent process along the sequence length in FlashAttention's backward pass, meaning longer sequences lead to greater error accumulation. Therefore, we maintain $\mathbf{dO}\mathbf{V}^\top$ in FP16 while accelerating the other four matrix multiplications using `INT8` per-block quantization. The algorithm for the forward is shown in Algorithm 3. Empirical results: Table 1 (c) shows the accuracy of the $\mathbf{dQ}$ with and without quantization of $\mathbf{dO}\mathbf{V}^\top$. We find that the accuracy of $\mathbf{dQ}$ is significantly improved when keeping $\mathbf{dO}\mathbf{V}^\top$ in FP16.

Table 1: Accuracy ablation using different quantization strategies.

| (a) Different FP4 choices | | | | (b) Different scale strategies for $\widetilde{\mathbf{P}}$ | | | | (c) Different data types for $\mathbf{dO}\mathbf{V}^\top$ | | | |
|---|---|---|---|---|---|---|---|---|---|---|---|
| Type | CosSim↑ | L1↓ | RMSE↓ | Method | CosSim | L1 | RMSE | Method | CosSim | L1 | RMSE |
| MXFP4 | 98.37% | 0.294 | 0.994 | Direct | 93.32% | 0.193 | 1.103 | INT8 | 97.47% | 0.171 | 2.440 |
| NVFP4 | **99.52%** | **0.077** | **0.201** | Two-level | **99.52%** | **0.077** | **0.201** | FP16 | **99.77%** | **0.039** | **0.692** |

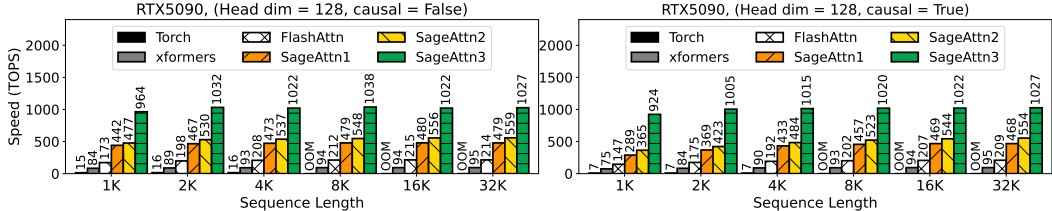

Figure 4: Speed comparison between `SageAttention3` and Baselines (`RTX5090`, headim=128).

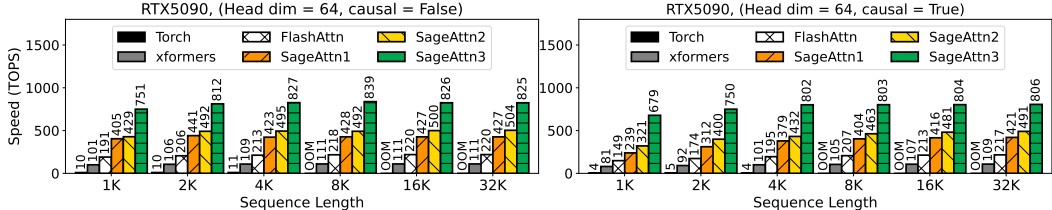

Figure 5: Speed comparison between `SageAttention3` and Baselines (`RTX5090`, headim=64).

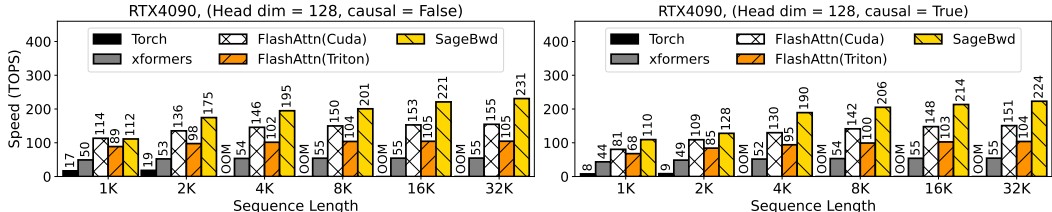

Figure 6: Speed comparison between `SageBwd` and Baselines (`RTX4090`, headim=128).

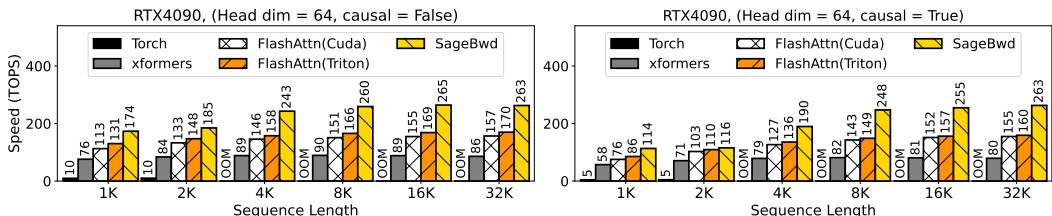

Figure 7: Speed comparison between `SageBwd` and Baselines (`RTX4090`, headim=64).

Table 2: End-to-end metrics comparison on various models.

| Model | Attention | CLIPSIM ↑ | CLIP-T ↑ | VQA-a ↑ | VQA-t ↑ | FScore ↑ |
|---|---|---|---|---|---|---|
| CogvideoX | Full-Precision (16bit) | 0.1865 | 0.9968 | 70.476 | 69.875 | 4.780 |
| | SageAttention2 (8bit) | 0.1880 | 0.9969 | 69.414 | 70.750 | 4.534 |
| | SageAttention3 (4bit) | **0.1881** | **0.9969** | **69.860** | **70.364** | **4.035** |
| Hunyuan Video | Full-Precision (16bit) | 0.1838 | 0.9993 | 68.998 | 78.891 | 1.4793 |
| | SageAttention2 (8bit) | 0.1836 | 0.9993 | 69.497 | 77.019 | 1.4741 |
| | SageAttention3 (4bit) | **0.1866** | **0.9993** | **70.552** | **75.440** | **1.232** |
| Mochi | Full-Precision (16bit) | 0.1828 | 0.9990 | 61.9840 | 61.0000 | 1.8042 |
| | SageAttention2 (8bit) | 0.1819 | 0.9990 | 61.0093 | 60.3732 | 1.7539 |
| | SageAttention3 (4bit) | **0.1800** | **0.9993** | **61.863** | **59.429** | **1.649** |

| Model | Attention | FID ↓ | sFID ↓ | CLIP ↑ | IR ↑ |
|---|---|---|---|---|---|
| Flux | Full-Precision (16bit) | 162.812 | 146.980 | 31.409 | 0.91 |
| | SageAttention2 (8bit) | 163.107 | 146.213 | 31.436 | 0.90 |
| | SageAttention3 (4bit) | **162.121** | **142.839** | **31.450** | **0.94** |
| Stable-Diffusion3.5 | Full-Precision (16bit) | 166.421 | 146.379 | 31.93 | 0.93 |
| | SageAttention2 (8bit) | 164.986 | 148.557 | 32.01 | 0.93 |
| | SageAttention3 (4bit) | **166.102** | **145.587** | **32.01** | **0.92** |

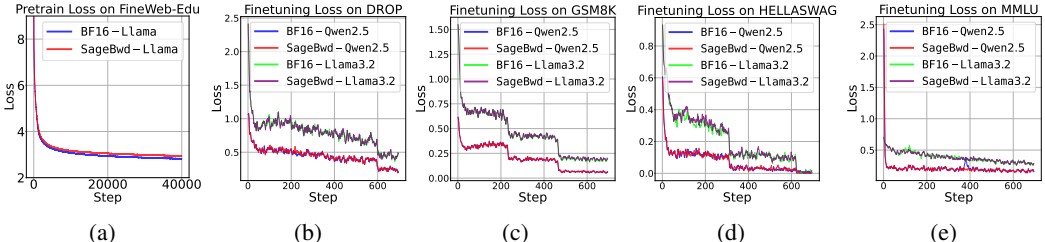

(a)     (b)     (c)     (d)     (e)

Figure 8: Pretraining and Finetuing loss curves of `BF16` and `8-bit` atttention.

Table 3: `8-bit` attention finetune results on `Qwen2.5` and `Llama3.2` models.

| Model | Method | GSM8K(Acc↑) | DROP(FI↑) | MMLU(Acc↑) | HELLASWAG(Acc↑) |
|---|---|---|---|---|---|
| Qwen2.5 (1.5B) | BF16 | 0.521 | 0.733 | 0.569 | 0.905 |
| | SageBwd | 0.520 | **0.734** | **0.574** | **0.911** |
| Qwen2.5 (3B) | BF16 | 0.601 | 0.785 | 0.640 | 0.944 |
| | SageBwd | **0.607** | 0.782 | **0.653** | 0.943 |
| Llama3.2 (1B) | BF16 | 0.259 | 0.641 | 0.464 | 0.828 |
| | SageBwd | **0.268** | 0.637 | 0.458 | 0.823 |

## 5 Experiments

**Main results.** `SageAttention3` is faster than FlashAttention and xformers by 5× and 11× on `RTX5090`, and maintains end-to-end metrics across various models. Furthermore, `SageBwd` is faster than FlashAttention and xformers by 1.67× and 3× on `RTX4090`, and achieves no measurable degradation in fine-tuning tasks.

### 5.1 Setup

**Models and attentions.** We validate the effectiveness of `SageAttention3` and `SageBwd` across a diverse set of representative models from language, image, and video generation. Specifically, we conduct experiments on: `Qwen2.5` [13] and `Llama3.2` [14] for text2text, `CogvideoX` [15], `HunyuanVideo` [16], and `Mochi` [17] for text2video, `Flux` [18], and `Stable-Diffusion3.5` [19] for text2image. We compare our method with FlashAttention2 [20], xformers [21], SageAttention [3], and SageAttention2 [4]. Please **note** that FlashAttention3 can only run on Hopper GPUs, so FlashAttention2 is already the fastest version for `RTX5090` and `RTX4090`.

**Datasets, metrics, and hyperparameters.** For the details about the datasets, metrics, and hyperparameters we used, please refer to Appendix A.3.

**Implementation.** We implement `SageAttention3` using CUTLASS [22] and CUDA, and implement `SageBwd` using OpenAI Triton [23].

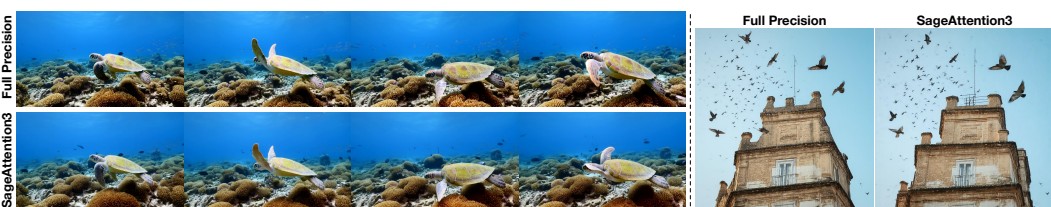

Figure 9: Visible examples of video generation on `HunyuanVideo` (left) and image generation on `Stable-Diffusion3.5` (right).

Table 4: End-to-end speedup performance using `SageAttention3` and `SageBwd`.

(a) Inference latency using `SageAttention3`.

| Model | Original | `Sage1` | `Sage2` | `Sage3` |
|---|---|---|---|---|
| `CogvideoX (2B)` | 64 s | 55 s | 46 s | **27 s** |
| `HunyuanVideo` | 489 s | 257 s | 240 s | **164 s** |

(b) One iteration training latency using `SageBwd`.

| Model | Original | `SageBwd` |
|---|---|---|
| `Llama (8K)` | 2.1 s | **1.9 s** |
| `Llama (16K)` | 6.0 s | **5.2 s** |

## 5.2 Efficiency and Effectiveness

**Kernel Speed.** Fig. 4 and 5 show the kernel speed of `SageAttention3` and baselines on RTX5090. We can see that `SageAttention3` achieves 4~$\mathbf{5}\times$ speedup over FlashAttention2 and 8~$\mathbf{11}\times$ speedup over xformers. Fig. 6 and 7 show the forward+backward speed of `SageBwd` and baselines on RTX4090. It shows that `SageBwd` achieves $\mathbf{1.67}\times$ speedup at most than FlashAttention2 and a higher speedup than FlashAttention2 implemented in Triton and xformers.

**End-to-end metrics loss of SageAttention3.** In Table 2, we compare the end-to-end quality metrics on various models using `SageAttention3` and other attention methods. The results demonstrate that `SageAttention3` almost incurs almost no end-to-end quality loss across these models.

**End-to-end metrics loss of SageBwd.** To evaluate the effectiveness of `SageBwd` on training tasks, we conduct two experiments. First, we fine-tune the base models of `Qwen2.5` (3B) and `Llama3.2` (1B) on GSM8K [24], DROP [25], MMLU [26], and HELLASWAG [27] datasets. Fig. 8 (b-e) shows the fine-tuning loss results, indicating that `SageBwd` perfectly aligns with `BF16`. Moreover, our evaluation of the fine-tuned models' answer quality across multiple test datasets (Table 3) demonstrates that `SageBwd` achieves the same performance as `BF16`. Second, we conduct pre-training tasks on FineWeb-Edu [28] using a `Llama` (400M) [29] model. Fig. 8 (a) shows the loss curve, indicating that while `SageBwd` can achieve loss convergence, its convergence speed is relatively slow. This limitation restricts its applicability in pretraining tasks.

**Visible example.** Fig. 9 visualizes some comparative examples of video generation on `HunyuanVideo` and image generation on `Stable-diffsion3.5` using `SageAttention3`. The results demonstrate that `SageAttention3` maintains full generation quality. Additional visible examples are provided in Fig. 10, 11, 13, and 14 in the Appendix.

**End-to-end speedup.** Table 4(a) and 4(b) summarize end-to-end inference and training latency improvements. The results show that `SageAttention3` (Table 4(a)) achieved about $\mathbf{3}\times$ (`HunyuanVideo`) and $\mathbf{2.4}\times$ (`CogVideoX`) end-to-end inference generation speedups on RTX5090. Furthermore, `SageBwd` (Table 4(b)) accelerates the training of Llama (1B) by about $\mathbf{1.15}\times$ using 8K/16K token micro-batches on RTX4090.

## 5.3 Benefit of Using Both SageAttention3 and SageBwd

Table 5: Comparison between `BF16` and `INT8` fine-tuning followed by `FP4` inference.

(a) `Qwen2.5-1.5B` results.

| Method | GSM8k ↑ | MMLU ↑ |
|---|---|---|
| BF16 Fine-tuning | 0.4912 | 0.4688 |
| `SageBwd` **Fine-tuning** | **0.5232** | **0.4934** |

(b) `Qwen2.5-3B` results.

| Method | GSM8k ↑ | MMLU ↑ |
|---|---|---|
| BF16 Fine-tuning | 0.5860 | 0.6000 |
| `SageBwd` **Fine-tuning** | **0.5945** | **0.6032** |

We first apply `SageBwd` during fine-tuning, followed by `SageAttention3` during inference. Specifically, we fine-tuned `Qwen2.5` for 1,000 steps using either `BF16` or `SageBwd`, and then evaluated inference performance using `SageAttention3`. The results on GSM8k and MMLU are shown in Table 5, `INT8` `SageBwd` fine-tuning followed by `FP4` `SageAttention3` inference achieves higher accuracy on GSM8k and MMLU, suggesting the approaches are complementary. This improvement is likely because `INT8` and `FP4` share a more similar representable data distribution, reducing the mismatch error compared to `BF16`.

### 5.4 INT8 SageBwd vs FP8 SageBwd

We choose `INT8` for `SageBwd` for two key reasons: (1) Higher gradient accuracy in attention backward. The backward of `INT8` attention yields more accurate gradients for $Q$, $K$, and $V$ compared to FP8. We evaluate all layers of `CogVideoX-2B` and report the L1 error and cosine similarity of the gradients in Table 6 and Table 7. For fairness, $\mathbf{dOV}^\top$ is kept in FP16 for both methods. As shown in the results, `INT8 SageBwd` achieves lower L1 error and higher cosine similarity than `FP8 SageBwd`. (2) Wider hardware support. `INT8` is supported on almost all modern GPUs, including NVIDIA `A100` and many non-NVIDIA devices (e.g., AMD `MI250` [30], Ascend `910B` [31]), while FP8 support remains limited to newer architectures. Additionally, we fine-tune `Qwen2.5-1.5B` and `Qwen2.5-3B` for 1,000 steps using either `INT8` or FP8 `SageBwd` (both with $\mathbf{dOV}^\top$ kept in FP16 for fairness), and then inference with `FP4 SageAttention3`. As shown in Table 8, models fine-tuned with `INT8` attention achieve higher accuracy on both GSM8K and MMLU benchmarks.

Table 6: L1 error of $Q$, $K$, and $V$ gradients.

| Method | $dQ \downarrow$ | $dK \downarrow$ | $dV \downarrow$ |
|---|---|---|---|
| INT8 SageBwd | **0.0290** | **0.0317** | **0.0423** |
| FP8 SageBwd | 0.0696 | 0.0999 | 0.0873 |

Table 7: Cos similarity of $Q$, $K$, and $V$ gradients.

| Method | $dQ \uparrow$ | $dK \uparrow$ | $dV \uparrow$ |
|---|---|---|---|
| INT8 SageBwd | **0.9987** | **0.9993** | **0.9995** |
| FP8 SageBwd | 0.9880 | 0.9910 | 0.9955 |

Table 8: Comparison of `INT8` and FP8 `SageBwd` fine-tuning on `Qwen2.5` models.

(a) `Qwen2.5-1.5B`

| Method | GSM8K $\uparrow$ | MMLU $\uparrow$ |
|---|---|---|
| INT8 Fine-tuning | **0.5232** | **0.4934** |
| FP8 Fine-tuning | 0.5031 | 0.4689 |

(b) `Qwen2.5-3B`

| Method | GSM8K $\uparrow$ | MMLU $\uparrow$ |
|---|---|---|
| INT8 Fine-tuning | **0.5945** | **0.6032** |
| FP8 Fine-tuning | 0.5868 | 0.5907 |

## 6 Related Work

Recent efficient attention works [2] that utilize hardware features to accelerate attention computation methods mainly include the following: FlashAttention [32] introduces tiling to reduce the GPU memory I/O between global memory and on-chip SRAM, achieving significant speedup. FlashAttention2 [20] improves the parallelism and warp partition strategies. FlashAttention3 [33] exclusively optimizes the kernel speed on the Hopper GPUs. xformers [21] accelerates attention using dedicated CUDA kernels. SageAttention [3] and SageAttention2 [4, 34] accelerate attention using quantization and some novel outlier smoothing techniques. RingAttention [35] extends FlashAttention to multi-GPU/Node environments. In these works, although FlashAttention3 proposes a version of FP8 attention, it has failed to be applied to video generation models in a plug-and-play way [4]. Moreover, the FP8 attention in FlashAttention3 does not support the backward pass, limiting its applicability to training tasks. Additionally, numerous efficient attention variants have emerged, including linear attention [36, 37, 38, 39, 40, 41] and sparse attention [42, 43, 44, 45, 46, 47, 48, 49, 50, 51, 52, 53, 54]. Although these works represent promising research directions, they are orthogonal to our work.

## 7 Conclusions

In this paper, we make two key contributions. Firstly, we design `SageAttention3`, the first microscaling FP4 attention for inference acceleration, achieving **1038** `TOPS` on `RTX5090`, which is a **5×** speedup than the fastest FlashAttention on `RTX5090`. Experiments show that `SageAttention3` could accelerate various models with no end-to-end quality metrics degradation. Secondly, we introduce the first trainable `8-bit` attention (`SageBwd`) for training acceleration and explore its feasibility in training tasks. We find that the `8-bit` attention could achieve lossless performance in fine-tuning tasks, but currently has some limitations in pertaining tasks.

**Future Work.** First, while `SageBwd` demonstrates faster performance than FP16 implementation, we observe a noticeable gap between its current speed and theoretical upper bounds. This gap may be caused by suboptimal Triton kernel implementations, which we plan to further optimize. Second, and more importantly, investigating the application of low-bit attention in pretraining tasks presents a promising research direction worthy of exploration.

## Acknowledgments

This work was supported by the NSFC Projects (Nos. 62550004, 92270001, 62376131). J.Z is also supported by the XPlorer Prize.

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

# A  Appendix

## A.1  Visible Comparison Examples

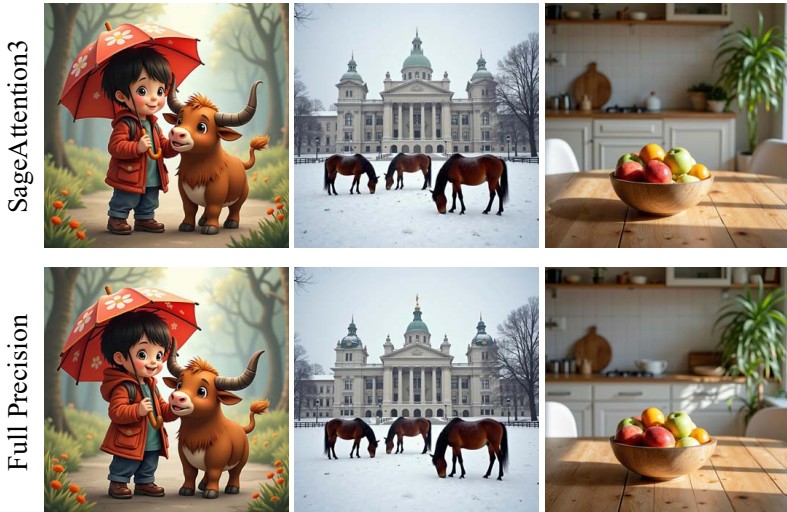

Figure 10: Visible examples of image generation on `Stable-Diffusion3.5`.

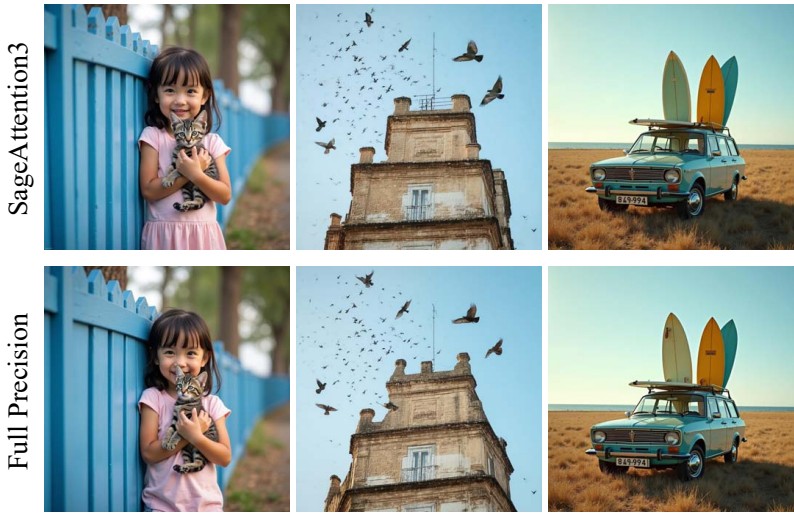

Figure 11: Visible examples of image generation on `Flux`.

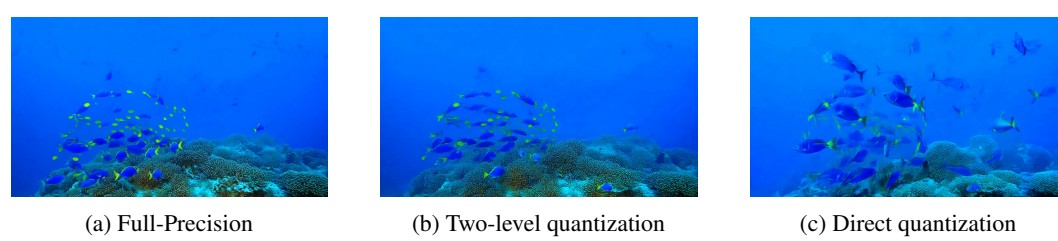

(a) Full-Precision     (b) Two-level quantization     (c) Direct quantization

Figure 12: Visual comparison of different scale strategies for $\widetilde{\mathbf{P}}$ from `CogVideoX`.

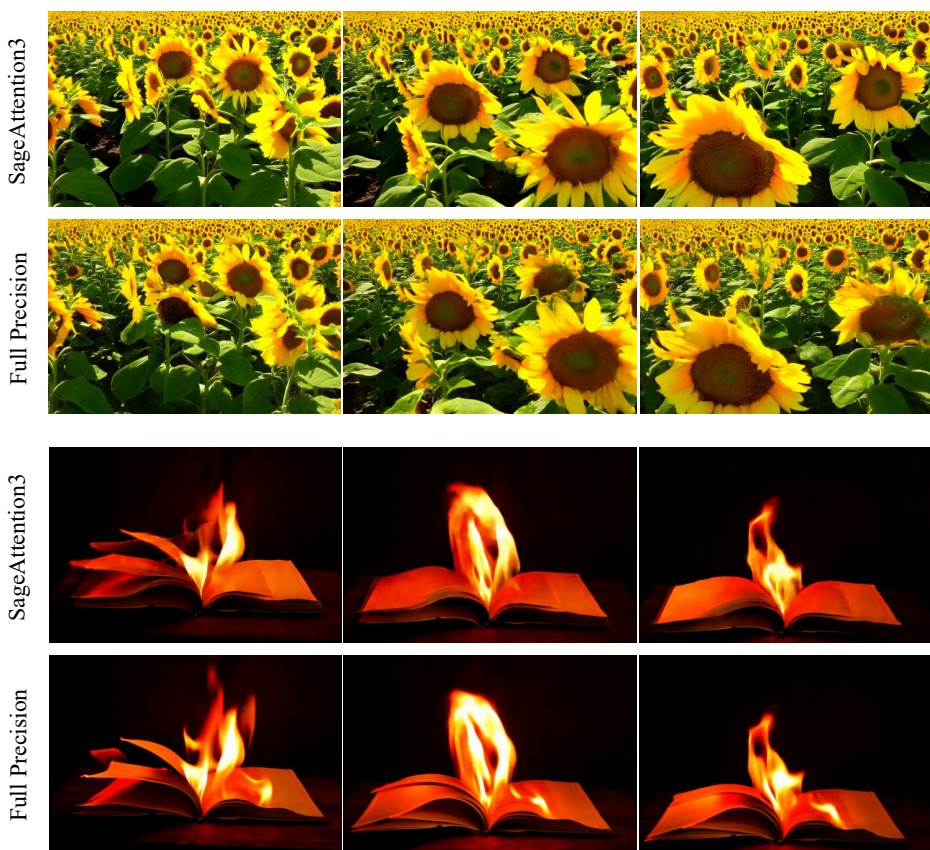

Figure 13: Visible examples of video generation on `CogVideoX`.

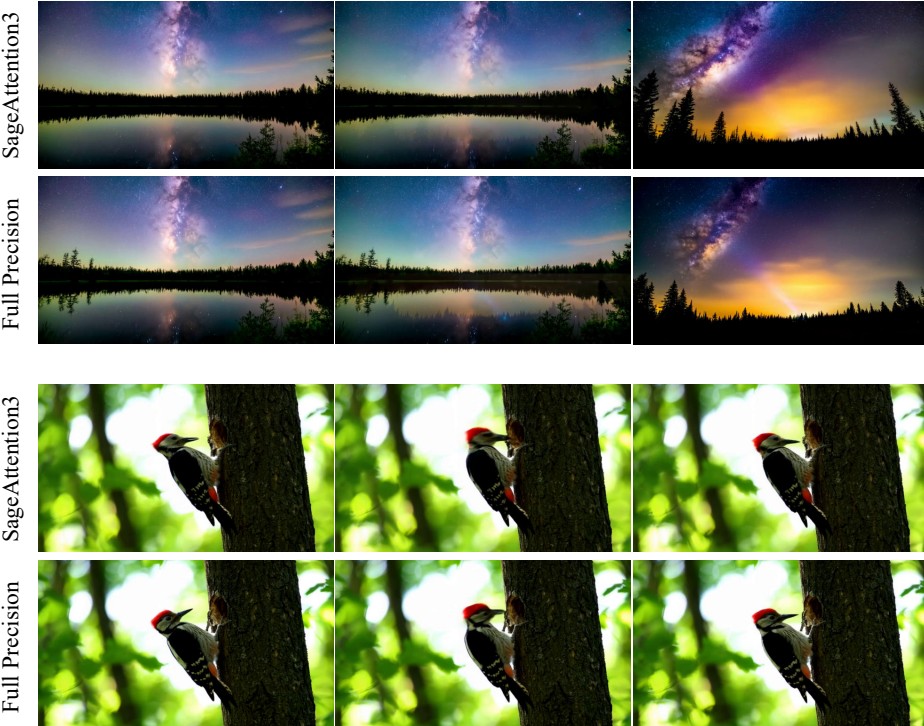

Figure 14: Visible examples of video generation on `HunyuanVideo`.

Fig. 10 and Fig. 11 show additional visual comparison examples of image generation tasks. Fig. 13 and Fig. 14 show more visual comparison examples of video generation tasks.

## A.2 Additional Kernel Speed Comparison

Fig. 15 and Fig. 16 show the forward kernel speed of `SageBwd`. Fig. 17 and Fig. 18 show the backward kernel speed of `SageBwd`. `SageBwd` achieved a **2x** speed up than FlashAttention in the forward propagation. `SageBwd` achieved a 1.2~**1.6x** speed up than FlashAttention in the backward propagation.

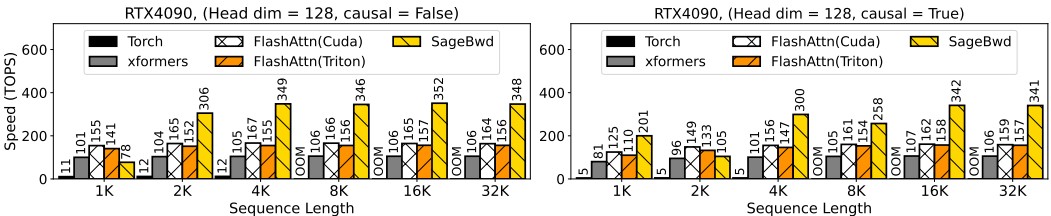

Figure 15: Forward speed comparison between `SageBwd` and Baselines (`RTX4090`, headim=128).

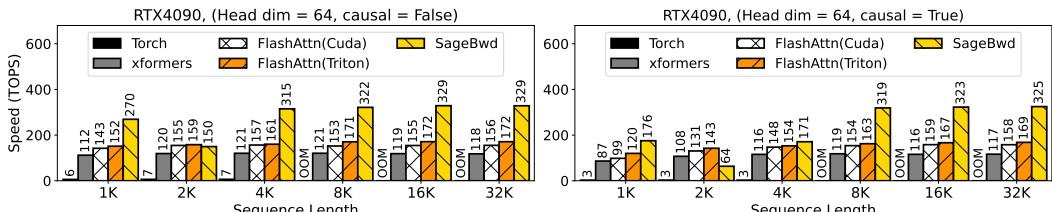

Figure 16: Forward speed comparison between `SageBwd` and Baselines (`RTX4090`, headim=64).

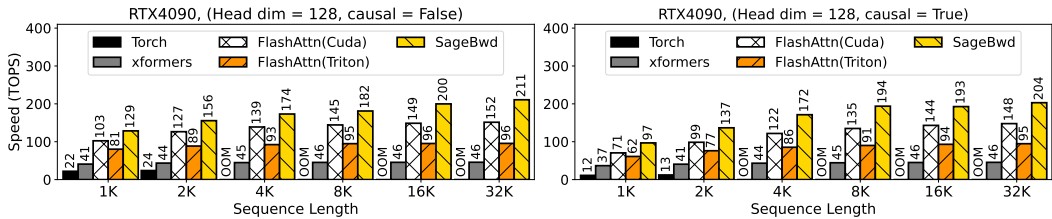

Figure 17: Backward speed comparison between `SageBwd` and Baselines (`RTX4090`, headim=128).

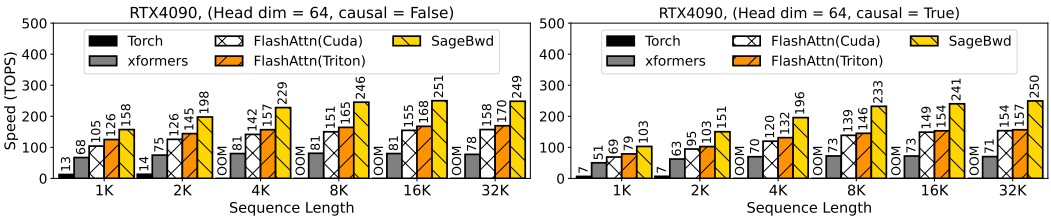

Figure 18: Backward speed comparison between `SageBwd` and Baselines (`RTX4090`, headim=64).

## A.3 Datasets, Metrics, and Hyperparameters

**Datasets.** Text-to-video models are evaluated using the open-sora [55] prompt sets. Text-to-image models are assessed on COCO annotations [56]. Language models are evaluated on GSM8K [24], DROP [25], MMLU [26], and HELLASWAG [27] datasets.

**End-to-end metrics.** For text-to-text models, we use Accuracy (Acc.) and F1-Score (F1). For text-to-video models, we evaluate the quality of generated videos on five metrics: CLIPSIM and CLIP-Temp (CLIP-T) [57] to measure the text-video alignment; (VQA-a) and (VQA-t) to assess the video aesthetic and technical quality, respectively; and Flow-score (FScore) for temporal consistency [58]. For text-to-image models, generated images are evaluated in three aspects: FID [59] and sFID [60] for fidelity evaluation, *Clipscore* (CLIP) [61] for text-image alignment, and *ImageReward* (IR) [62] for human preference.

**Accuracy metrics.** We use three metrics to assess the accuracy of quantized attention output $O'$ compared to attention output in full-precision $O$: First, we flatten $O'$ and $O$ into vectors in the shape of $1 \times n$. Then, Cosine similarity: $CosSim = \sum OO'/\sqrt{\sum O^2}\sqrt{\sum O'^2}$, Relative L1 distance: $L1 = \sum |O - O'|/\sum |O|$, Root mean square error: $RMSE = \sqrt{(1/n)\sum(O - O')^2}$.

**Hyperparameters.** For pretraining tasks, we use a 400M model with a hidden size of 1024, 20 layers, an intermediate size of 3072, and 16 attention heads. The training uses a learning rate of 1e-3 with linear decay over 1000 warmup steps, and each step processes 2M tokens. For finetuning tasks, we train for 700 steps using a learning rate of 3e-5 with linear decay and 100 warmup steps with a batch size of 32 on GSM8K dataset and 128 on MMLU, DROP, and HELLASWAG datasets.

Figure 19: FP4 operand A register layout - rows 0 and 8, thread 0-3, entries 0-15.

Figure 20: FP32 accumulator register layout - rows 0 and 8, thread 0-3, entries 0-15.

Figure 21: Permuted FP32 accumulator register layout - rows 0 and 8, thread 0-3, entries 0-15.

## A.4 Additional Experiments of using `SageBwd`

Table 9–14 show `Qwen2.5` (1.5B), `Qwen2.5` (3B), and `Llama3.2` (3B) fine-tuning results on four datasets with five different random seeds. The average and standard deviation show `SageBwd` is highly consistent with `BF16` across various random seeds.

Table 9: Comparison of `SageBwd` and `BF16` performance on GSM8K and DROP across different seeds on `Qwen2.5` (1.5B).

| Seed | GSM8K | | DROP | |
|---|---|---|---|---|
| | SageBwd | BF16 | SageBwd | BF16 |
| 42 | 0.5133 | 0.5125 | 0.7316 | 0.7364 |
| 233 | 0.5027 | 0.5042 | 0.7269 | 0.7295 |
| 1234 | 0.4973 | 0.4973 | 0.7329 | 0.7342 |
| 5678 | 0.5201 | 0.5208 | 0.7340 | 0.7332 |
| 1 | 0.5049 | 0.5057 | 0.7278 | 0.7404 |
| **Avg** | 0.5077 | 0.5081 | 0.7307 | 0.7348 |
| **Std** | 0.0090 | 0.0089 | 0.0032 | 0.0040 |

Table 10: Comparison of `SageBwd` and `BF16` performance on MMLU and HellaSwag across different seeds on `Qwen2.5` (1.5B).

| Seed | MMLU | | HellaSwag | |
|---|---|---|---|---|
| | SageBwd | BF16 | SageBwd | BF16 |
| 42 | 0.5814 | 0.5873 | 0.9089 | 0.9065 |
| 233 | 0.5746 | 0.5785 | 0.9082 | 0.9049 |
| 1234 | 0.5805 | 0.5836 | 0.9025 | 0.9047 |
| 5678 | 0.5736 | 0.5693 | 0.9112 | 0.9053 |
| 1 | 0.5830 | 0.5823 | 0.9058 | 0.9075 |
| **Avg** | 0.5786 | 0.5802 | 0.9073 | 0.9058 |
| **Std** | 0.0043 | 0.0069 | 0.0033 | 0.0012 |

Table 11: Comparison of `SageBwd` and `BF16` performance on GSM8K and DROP across different seeds on `Qwen2.5` (3B).

| Seed | GSM8K | | DROP | |
|---|---|---|---|---|
| | SageBwd | BF16 | SageBwd | BF16 |
| 42 | 0.5982 | 0.6232 | 0.7800 | 0.7812 |
| 233 | 0.5997 | 0.5974 | 0.7786 | 0.7812 |
| 1234 | 0.6156 | 0.6103 | 0.7786 | 0.7824 |
| 5678 | 0.6065 | 0.6012 | 0.7816 | 0.7853 |
| 1 | 0.6171 | 0.6073 | 0.7813 | 0.7832 |
| **Avg** | 0.6074 | 0.6079 | 0.7800 | 0.7827 |
| **Std** | 0.0001 | 0.0001 | 0.0000 | 0.0000 |

Table 12: Comparison of `SageBwd` and `BF16` performance on MMLU and HellaSwag across different seeds on `Qwen2.5` (3B).

| Seed | MMLU | | HellaSwag | |
|---|---|---|---|---|
| | SageBwd | BF16 | SageBwd | BF16 |
| 42 | 0.6434 | 0.6425 | 0.9419 | 0.9402 |
| 233 | 0.6431 | 0.6437 | 0.9405 | 0.9402 |
| 1234 | 0.6492 | 0.6492 | 0.9414 | 0.9429 |
| 5678 | 0.6531 | 0.6400 | 0.9430 | 0.9440 |
| 1 | 0.6510 | 0.6454 | 0.9446 | 0.9434 |
| **Avg** | 0.6480 | 0.6442 | 0.9423 | 0.9421 |
| **Std** | 0.0000 | 0.0000 | 0.0000 | 0.0000 |

Table 13: Comparison of `SageBwd` and `BF16` performance on GSM8K and DROP across different seeds on `Llama3.2` (1B).

| Seed | GSM8K | | DROP | |
| --- | --- | --- | --- | --- |
| | SageBwd | BF16 | SageBwd | BF16 |
| 42 | 0.2722 | 0.2547 | 0.6367 | 0.6447 |
| 233 | 0.2661 | 0.2570 | 0.6456 | 0.6424 |
| 1234 | 0.2616 | 0.2873 | 0.6439 | 0.6352 |
| 5678 | 0.2684 | 0.2585 | 0.6372 | 0.6409 |
| 1 | 0.2646 | 0.2335 | 0.6393 | 0.6441 |
| **Avg** | 0.2666 | 0.2582 | 0.6405 | 0.6414 |
| **Std** | 0.0000 | 0.0003 | 0.0000 | 0.0000 |

Table 14: Comparison of `SageBwd` and `BF16` performance on MMLU and HellaSwag across different seeds on `Llama3.2` (3B).

| Seed | MMLU | | HellaSwag | |
| --- | --- | --- | --- | --- |
| | SageBwd | BF16 | SageBwd | BF16 |
| 42 | 0.4665 | 0.4705 | 0.8230 | 0.8319 |
| 233 | 0.4646 | 0.4560 | 0.8327 | 0.8256 |
| 1234 | 0.4702 | 0.4757 | 0.8202 | 0.8243 |
| 5678 | 0.4580 | 0.4639 | 0.8232 | 0.8276 |
| 1 | 0.4666 | 0.4691 | 0.8218 | 0.8236 |
| **Avg** | 0.4652 | 0.4670 | 0.8242 | 0.8266 |
| **Std** | 0.0000 | 0.0000 | 0.0000 | 0.0000 |

## A.5 Transposing $V$.

Performing the forward propagation of attention in full `FP4` precision poses additional challenges compared to `FP16`. The input tensors $Q$, $K$, and $V$ are typically contiguous in the head dimensions. However, the row-major constraints on `FP4` MMA for the second GEMM necessitate $V$ to be contiguous in the sequence length dimension. Calling a standalone pre-processing transpose kernel for this purpose incurs excessive overhead, particularly during inference, which is often a memory-bound situation. We address the problem by kernel fusion. For the first problem, we fuse the transpose of $V$ into the quantization kernel, thereby avoiding additional I/O overhead.

## A.6 Accmulated Quantization Error Analysis.

Table 15: Layer-wise L1 error analysis of `SageAttention3` on `CogVideoX-2B`. The second row shows the results by retaining the three most sensitive layers in FP16.

| Method | Layer1 ↓ | Layer10 ↓ | Layer20 ↓ | Layer30 ↓ |
| --- | --- | --- | --- | --- |
| Use `SageAttention3` directly | 0.0076 | 0.0922 | 0.1146 | 0.0571 |
| Keep 3 most sensitive layers in FP16 | 0.0076 | **0.0447** | **0.0773** | **0.0429** |

To explore the issue of accumulated quantization error across layers, we conduct an analysis using `SageAttention3` on `CogVideoX-2B` and report the per-layer L1 error in Table 15. We observe that the accumulated error generally increases with layer depth, though it occasionally decreases in deeper layers, suggesting partial error cancellation. To mitigate this drift, we apply a simple yet effective strategy: keeping the three layers with the largest observed error growth in `FP16` precision. As shown in the table, this adjustment significantly reduces the overall error accumulation across layers.

## A.7 Ablation of Smoothing Techniques.

Table 16: Ablation of attention accuracy with different smoothing methods on `CogVideoX-2B`. Smoothing K and Smoothing Q are techniques from `SageAttention` and `SageAttention2`.

| Method | Cossim ↑ | L1 Error ↓ | RMSE ↓ |
|---|---|---|---|
| None | 0.915642 | 0.335867 | 0.303483 |
| SmoothQuant | 0.930125 | 0.267617 | 0.252883 |
| Hadamard | 0.941222 | 0.262047 | 0.223970 |
| **Smoothing_Q** | **0.982848** | **0.115658** | **0.125862** |
| **Smoothing_K** | **0.991176** | **0.094832** | **0.097668** |

To investigate the impact of different smoothing strategies on attention accuracy, we compare several existing techniques, including SmoothQuant [63] and Hadamard transformations, which provide per-token or per-tensor scaling control. However, we find these methods less effective in our setting. `SageAttention3` inherits the smoothing Q and smoothing K mechanisms introduced in `SageAttention2`. We conduct an ablation study on all layers of `CogVideoX-2B` to evaluate their effects. As shown in Table 16, both smoothing Q and smoothing K yield substantially higher cosine similarity and lower reconstruction errors, demonstrating their effectiveness in stabilizing quantized attention computation.

## A.8 Theoretical Speed Comparison.

Table 17: Theoretical throughput comparison between FlashAttention3 and `SageAttention3` across different GPUs.

| Method | B300 TOPS ↑ | B200 TOPS ↑ | RTX5090 TOPS ↑ |
|---|---|---|---|
| FlashAttention3 | 2500 | 2500 | 209.5 |
| FlashAttenion3 (FP8) | 5000 | 5000 | 419 |
| SageAttention3 (FP4) | **15000** | **10000** | **1676** |

To provide a theoretical comparison with FlashAttention3, we refer to NVIDIA's official documentation on throughput (TOPS) across different precisions. Since FlashAttention3 is currently only supported on `H100` GPUs, a direct empirical comparison is not feasible. Instead, we estimate the theoretical compute throughput of both FlashAttention3 and our `SageAttention3` on GPUs that support FP4 Tensor Cores (`B300`, `B200`, and `RTX5090`). As summarized in Table 17, `SageAttention3` achieves substantially higher theoretical peak throughput, highlighting its potential for further accelerating attention computation beyond FlashAttention-3.

## A.9 FlashAttentions vs SageAttentions.

Table 18: Speed–accuracy trade-off of different attention methods.

| Method | TOPS on 5090 ↑ | TOPS on H100 ↑ | Accuracy (CosSim) ↑ |
|---|---|---|---|
| FlashAttention2 | 214 | 338 | 100.000% |
| FlashAttention3 (16bit) | N/A | 470 | 100.000% |
| FlashAttention3 (8bit) | N/A | 890 | 98.570% |
| SageAttention1 | 479 | 518 | 99.996% |
| SageAttention2 (8bit) | 643 | 885 | 99.995% |
| SageAttention3 (4bit) | **1038** | N/A | **99.551%** |

To illustrate the trade-off between accuracy and speed, we recorded the accuracy (Cosine similarity) of various attention methods across all layers of `CogVideoX-2B`, along with their theoretical throughput on `RTX5090` and `H100` GPUs. These results are summarized in the Table 18.

## A.10 Analysis of Two-Level Quantization.

*Proof.* We analyze the relative quantization error of $\widetilde{\mathbf{P}}$ using both `direct quantization` and `two-level quantization` as follows:

For `direct quantization`, the relative quantization error, denoted as $E_1$, is defined as:

$$\mathbf{s_P}, \hat{\mathbf{P}} = \phi(\widetilde{\mathbf{P}}), \quad E_1 = \frac{|\mathbf{s_P} \times \hat{\mathbf{P}} - \widetilde{\mathbf{P}}|}{|\widetilde{\mathbf{P}}|} \tag{9}$$

For `two-level quantization`, the first level proceeds as:

$$\mathbf{s_{P_1}} = \mathrm{rowmax}(\widetilde{\mathbf{P}})/(448 \times 6), \quad \widetilde{\mathbf{P}}_2 = \widetilde{\mathbf{P}}/\mathbf{s_{P_1}},$$

The quantization error introduced in this first step is negligible because $\widetilde{\mathbf{P}}$, $\widetilde{\mathbf{P}}_2$, and $s_{P_1}$ are all represented in FP32 format.

We focus primarily on the second-level quantization, where the relative quantization error $E_2$ is given by:

$$\mathbf{s_{P_2}}, \hat{\mathbf{P}}_2 = \phi(\widetilde{\mathbf{P}}_2), \quad E_2 = \frac{|\mathbf{s_{P_2}} \times \hat{\mathbf{P}}_2 - \widetilde{\mathbf{P}}_2|}{|\widetilde{\mathbf{P}}_2|} \tag{10}$$

The key difference between Equation 9 and 10 lies in the range of the FP8 scale factor.

Let $\{X\}_n$ denote the number of distinct representable values in the set $X$. Then:

in `direct quantization`:

$$0 \leq \mathbf{s_P} \leq 0.167, \quad \mathbf{s_P} \in \mathbf{E4M3}, \quad \{\mathbf{s_P}\}_n = 35$$

In `two-level quantization`:

$$0 \leq \mathbf{s_{P_2}} \leq 448.0, \quad \mathbf{s_{P_2}} \in \mathbf{E4M3}, \quad \{\mathbf{s_{P_2}}\}_n = 127$$

Since $\{\mathbf{E2M1}\}_n = 8$, the number of unique outputs after dequantization is:

For `direct quantization`:

$$\widetilde{\mathbf{P}}' = \mathbf{s_P} \times \hat{\mathbf{P}}, \quad \{\widetilde{\mathbf{P}}'\}_n = 35 \times 8 = 280$$

For `two-level quantization`:

$$\widetilde{\mathbf{P}}_2' = \mathbf{s_{P_2}} \times \hat{\mathbf{P}}_2, \quad \{\widetilde{\mathbf{P}}_2'\}_n = 127 \times 8 = 1016$$

Let $\Delta(p_i)$ denote the interval between the two nearest quantization levels surrounding the value $p_i \in \widetilde{\mathbf{P}}$. Then the absolute quantization error satisfies:

$$|\hat{p}_i - p_i| \leq \frac{\Delta(p_i)}{2}$$

The relative error $\varepsilon$ satisfies:

$$\varepsilon_i \leq \frac{\Delta(p_i)}{2 \times p_i}$$

Given that $\{\widetilde{\mathbf{P}}_2'\}_n > \{\widetilde{\mathbf{P}}'\}_n$, the quantization intervals in the two-level scheme are finer:

$$\frac{\Delta(\widetilde{\mathbf{P}}_2)}{\widetilde{\mathbf{P}}_2} < \frac{\Delta(\widetilde{\mathbf{P}})}{\widetilde{\mathbf{P}}}$$

Thus, the relative quantization error satisfies:

$$\frac{|\widetilde{\mathbf{P}}_2' - \widetilde{\mathbf{P}}_2|}{\widetilde{\mathbf{P}}_2} < \frac{|\mathbf{P}' - \widetilde{\mathbf{P}}|}{\widetilde{\mathbf{P}}}$$

Which leads to the conclusion:

$$E_2 < E_1$$

$\square$

## A.11 Analysis of the Benefit of Keeping $\mathbf{dO}_i \mathbf{V}_j^\top$ in FP16 in `SageBwd`.

The backward pass of `SageBwd` involves 5 MatMuls. The accuracy of $\mathbf{S}_{ij} = \mathbf{Q}_i \mathbf{K}_j^\top$ is fully addressed in SageAttention2. The remaining four are as follows:

(1) $\mathbf{dP}_{ij} = \mathbf{dO}_i \mathbf{V}_j^\top$.

(2) $\mathbf{dQ}_i \leftarrow \mathbf{dQ}_i + \mathbf{dS}_{ij} \mathbf{K}_j$

(3) $\mathbf{dK}_j \leftarrow \mathbf{dK}_j + \mathbf{dS}_{ij}^\top \mathbf{Q}_i$

(4) $\mathbf{dV}_j \leftarrow \mathbf{dV}_j + \mathbf{P}_{ij}^\top \mathbf{dO}_i$

We choose to keep (1) in FP16, while quantizing others to INT8. This choice can be formally justified:

*Proof.* Following [64], we assume that any matrix $\mathbf{X} \in \mathbb{R}^{n \times d}$ (e.g. $\mathbf{Q}, \mathbf{K}, \mathbf{V}, \mathbf{dO}$) satisfies:

- The entries in $\mathbf{X}$ are mutually independent.

- $\mathbf{X}_{ij} \sim N(\mu_{\mathbf{X},j}, \sigma_{\mathbf{X},j}^2)$, i.e. the distribution of each token is identical.

The quantization error of a matrix $\mathbf{X}$ is denoted as:

$$\Delta \mathbf{X} := s_{\mathbf{X}} \hat{\mathbf{X}} - \mathbf{X}, \quad \text{where } s_{\mathbf{X}}, \hat{\mathbf{X}} = \psi(\mathbf{X}).$$

For example, consider the error in $\mathbf{dQ}$. Neglecting second-order error terms, we have:

$$\Delta \mathbf{dQ} = \underbrace{(\mathbf{P} \circ (\mathbf{dO}\Delta\mathbf{V}^\top + \Delta\mathbf{dO}\mathbf{V}^\top))\mathbf{K}}_{\Delta \mathbf{dQ}^{(1)} \text{ from (1)}} + \underbrace{\Delta \mathbf{dS}\mathbf{K} + \mathbf{dS}\Delta\mathbf{K}}_{\mathbf{dQ}^{(2)} \text{ from (2)}}.$$

Here, $\mathbf{dS} = \mathbf{P} \circ (\mathbf{dP} - D) = \mathbf{P} \circ (\mathbf{dO}\mathbf{V}^\top - D)$, where $D = \mathbf{dO} \odot \mathbf{O}$. In element-wise terms (the subscript denotes a single element):

$$\mathbf{dS}_{ij} = \mathbf{P}_{ij} \sum_k \mathbf{dO}_{ik}(\mathbf{V}_{jk} - \mathbf{O}_{ik}) = \mathbf{P}_{ij} \sum_k \mathbf{dO}_{ik} \left( \mathbf{V}_{jk} - \sum_\ell \mathbf{P}_{i\ell} \mathbf{V}_{\ell k} \right)$$

Since $\mathbf{V}$ is independent of other variables, by linearity of expectation:

$$\mathbb{E}[\mathbf{dS}_{ij}] = \mathbb{E}\left[ \mathbf{P}_{ij} \sum_k \mathbf{dO}_{ik} \left( \mu_{\mathbf{V},k} - \sum_\ell \mathbf{P}_{i\ell} \mu_{\mathbf{V},k} \right) \right] = 0.$$

Moreover, as negating $\mathbf{V}$ flips the sign of $\mathbf{dS}_{ij}$, the PDF of $\mathbf{dS}_{ij}$ is symmetric. Using a "round-to-nearest" quantization policy, we have $\mathbb{E}[\Delta \mathbf{dS}] = 0$. Thus

$$\mathbb{E}\left[ \mathbf{dQ}^{(2)} \right] = \mathbb{E}[\Delta \mathbf{dS}\mathbf{K} + \mathbf{dS}\Delta\mathbf{K}] = 0,$$

while $\mathbb{E}\left[ \Delta \mathbf{dQ}^{(1)} \right]$ is generally non-zero (e.g. when distributions have non-zero means), indicating that $\mathbf{dQ}$'s error is dominated by $\Delta \mathbf{dQ}^{(1)}$.

$\square$

## A.12 Broader Impact

This paper presents work that aims to advance the field of efficient machine learning systems. It can be used to accelerate the inference and training processes of various models. None of the negative impacts we feel must be specifically highlighted here.

