# OpenReview forum: "SageAttention3: Microscaling FP4 Attention for Inference and An Exploration of 8-Bit Training"
_NeurIPS.cc/2025/Conference — NeurIPS 2025 spotlight_

### Official Review · Reviewer_wcep · 2025-07-02

**Clarity:** 3
**Significance:** 3
**Originality:** 3
**Rating:** 5
**Confidence:** 5

**Summary:**

This paper proposes SageAttention3, which introduces two major advances for efficient attention inference and training. First, it uses a new NVFP4 attention approach that leverages FP4 Tensor Cores on NVIDIA Blackwell GPUs for better acceleration. The method uses a two-level scaling scheme to quantize the attention map,  fully utilizing the presentative range of the FP8 scaling factor. Combined with the 1×16 group-wise microscaling NVFP4 quantization, it achieves up to 5× speedup over FlashAttention2 on RTX5090 GPUs, with no noticeable drop across large language, image, and video generation models. Second, this paper proposes SageBwd, an 8-bit trainable attention that can accelerate both the forward and backward for training. Experiments show that it can achieve lossless performance in fine-tuning tasks, but with slower convergence in pretraining. Together, these features significantly improve the efficiency of model inference and training on modern hardware.

**Questions:**

- For the 8-bit attention for training, why int8 and not fp8? FP8 training is already widely used in LLM training, e.g., DeepSeek-V3. If the attention operator could also use FP8 for training, it could further boost the model training efficiency, with almost all of the computation being performed in 8-bit.

**Ethical Concerns:**

["NO or VERY MINOR ethics concerns only"]

**Final Justification:**

Most of my questions have been addressed, so I raised my rating.

**Limitations:**

yes

**Paper Formatting Concerns:**

No paper formatting concern.

**Quality:**

3

**Strengths And Weaknesses:**

- Strengths
	- This paper is well-written and easy to follow.
	- The proposed method combines new hardware features and proposes NVFP4 attention, SageAttention3, for better model efficiency. SageAttention3 can achieve impressive speedups across different model architectures and benchmarks.
	- Experiments show real speedups on GPUs, demonstrating the effectiveness of the proposed method.
- Weakness
	- The two contributions, nvfp4 microscaling quantization for inference efficiency and int8 quantization for training efficiency, are not linked. It would be better to have another session to show how to fine-tune models with int8 and then convert them to NVFP4 for better inference efficiency. In this way, those two can be tied together.
	- There are some other 4-bit quantization methods like QServe[1], Atom[2], and QuaRot[3]. While they don't support 4-bit computation in attention, 4-bit memory access can still boost model performance, especially for memory-bounded scenarios. So it would be good to compare SageAttention3 with other 4-bit quantization methods to demonstrate the advantages of SageAttention3.

[1] Yujun Lin, et al. QServe: W4A8KV4 Quantization and System Co-design for Efficient LLM Serving. MLSys, 2025.

[2] Yilong Zhao, et al. Atom: Low-bit Quantization for Efficient and Accurate LLM Serving. MLSys, 2024.

[3] Saleh Ashkboos, et al. QuaRot: Outlier-Free 4-Bit Inference in Rotated LLMs. NeurIPS, 2024.

---

> ### Author Rebuttal · Authors · 2025-07-30
>
> Dear Reviewer wcep,
> Thank you for your valuable suggestions and questions. Below, we address each point raised.
>
> ---
> **W1.** The two contributions, nvfp4 microscaling quantization for inference efficiency and int8 quantization for training efficiency, are not linked. It would be better to have another session to show how to fine-tune models with int8 and then convert them to NVFP4 for better inference efficiency. In this way, those two can be tied together.
>
> **Response:** Thank you for your insightful and important suggestion. We are very willing to add a Section to show how to fine-tune models with int8 and then convert them to NVFP4 for better efficiency and accuracy. We add an experiment that combines them: first applying INT8 attention during fine-tuning, followed by FP4 attention during inference. Specifically, we fine-tuned Qwen2.5 for 1,000 steps using either BF16 or our INT8 attention, and then evaluated inference performance using our FP4 attention. The results on GSM8k and MMLU are shown below, where INT8 fine-tuning followed by FP4 inference yields higher accuracy, indicating that the two methods can be complementary, and their combination can deliver additional benefits.
>
> **Table 1: Comparison on Qwen2.5-1.5B.**
>
> | All using FP4 Inference|Gsm8k $\uparrow$|Mmlu $\uparrow$ |
> | :-----------------:|:--------------:|:-------------: |
> | BF16 Fine-tuning|0.4912| 0.4688 |
> | INT8 Fine-tuning|**0.5232**| **0.4934** |
>
> **Table 2: Comparison on Qwen2.5-3B.**
>
> | All using FP4 Inference|Gsm8k $\uparrow$|Mmlu $\uparrow$ |
> | :---------------------:|:--------------:|:-------------: |
> | BF16 Fine-tuning | 0.5860 | 0.6000 |
> | INT8 Fine-tuning | **0.5945** | **0.6032** |
>
> This improvement is likely because INT8 and FP4 share a more similar representable data distribution, reducing the mismatch error compared to BF16.
>
>
> ---
> **W2.** There are some other 4-bit quantization methods like QServe[1], Atom[2], and QuaRot[3]. While they don't support 4-bit computation in attention, 4-bit memory access can still boost model performance, especially for memory-bounded scenarios. So it would be good to compare SageAttention3 with other 4-bit quantization methods to demonstrate the advantages of SageAttention3. (QServe, Atom, QuaRot)
>
> **Response:** Thank you for your very professional and important suggestion. Yes, as you pointed out, the methods you mentioned do not quantize $Q$, $K$, and $V$ entirely to 4-bit, nor do they perform attention computations at FP4 precision. Our work is the first to leverage hardware FP4 Tensor Cores to perform the entire attention computation in FP4, directly accelerating the computation. Strictly speaking, these approaches are not directly comparable, similar to how FP8-quantized FlashAttention3 is not compared against KV-cache-only quantization methods.
>
> To provide a more complete evaluation, we add two sets of comparison experiments. First, we compare SageAttention3 with QServe and Atom (please note that GPTQ is irrelevant here as it does not modify attention) on Qwen2.5-1B. As shown in the Table below, SageAttention3 achieves the best accuracy while also providing the fastest pre-filling speed. For decoding speed, because Atom and QServe cannot be compiled on the RTX 5090, we report their theoretical decoding speedup.
>
> **Table 3: Comparison on Qwen2.5.**
>
> | Method |wikitext PPL $\downarrow$|LAMBDA $\uparrow$|Prefilling(64K) Latency $\downarrow$|Theoretical decoding speedup $\uparrow$ |
> | ---------|------------------------:|----------------:|-----------------------------------:|--------------------------------------: |
> | Qserve | 10.90296 | 0.631| 10.20257 | 4x |
> | Atom | 10.89669 | 0.631| 10.22090 | 4x |
> | SageAttn3| **9.56827** | **0.726**| **6.09898** | 4x |
>
> Second, we compare SageAttention3 and QServe on CogVideoX-2B. As shown in the Table below, SageAttention3 not only achieves higher accuracy but also provides substantial speedup.
>
> **Table 4: Comparison on CogVideoX.**
>
> | Method |CLIPSIM ↑|VQA-a ↑|VQA-t ↑|FScore ↑|Latency $\downarrow$ |
> | ---------|--------:|------:|------:|-------:|-------------------: |
> | Qserve | 0.1683|19.7863|49.4831| 2.4488| 65s |
> | Atom | 0.1706|20.0896|46.9320| 2.4024| 64s |
> | SageAttn3|**0.1881**|**69.8600**|**70.3640**|**4.0350**| **27s** |
>
>
>
> ---
> **Q1.** For the 8-bit attention for training, why int8 and not fp8? FP8 training is already widely used in LLM training, e.g., DeepSeek-V3. If the attention operator could also use FP8 for training, it could further boost the model training efficiency, with almost all of the computation being performed in 8-bit.
>
> **Response:** Thank you for your highly professional and valuable question. Many works have explored INT8 training for linear layers, and FP8 has often been chosen in practice due to its better end-to-end accuracy. Our choice of INT8 for attention training is based on two key reasons:
>
> 1. **Higher gradient accuracy in attention backward:** The backward pass of INT8 attention yields more accurate gradients for $Q$, $K$, and $V$. As shown in the tables below, evaluating all layers of CogVideoX-2B demonstrates that INT8 SageBwd achieves lower L1 error and higher cosine similarity than FP8 SageBwd (with $\mathbf{dOV}^\top$ kept in FP16 for fairness; the only difference is the data type).
>
> **Table 5: L1 Error of Q, K, V gradient.**
>
> | Method|$dQ$ L1 Error $\downarrow$ |$dK$ L1 Error $\downarrow$|$dV$ L1 Error $\downarrow$ |
> | :---------:|:---------:|:---------:|------ |
> | INT8 SageBwd|**0.0290**|**0.0317**|**0.0423** |
> | FP8 SageBwd|0.0696|0.0999|0.0873 |
>
> **Table 6: Cossim accuracy of Q, K, V gradient.**
>
> | Method|$dQ$ Cossim $\uparrow$|$dK$ Cossim $\uparrow$|$dV$ Cossim $\uparrow$ |
> | :-------:|:-------:|:-------:|------ |
> | INT8 SageBwd|0.9987|0.9993|0.9995 |
> | FP8 SageBwd|0.9880|0.9910|0.9955 |
>
> 2. **Wider hardware support:** INT8 is supported on virtually all modern GPUs, including A100 and many non-NVIDIA GPUs (e.g., AMD MI250, Ascend 910B), while FP8 support is still limited to newer architectures.
>
> Additionally, we compare the end-to-end performance of INT8 and FP8 trainable attention when combined with FP4 SageAttention3 for inference. Specifically, we fine-tuned Qwen2.5-1.5B and Qwen2.5-3B for 1,000 steps using either INT8 or FP8 SageBwd (both with $\bf dOV^\top$ kept in FP16 for fairness), then evaluated inference with FP4 SageAttention3. As shown below, models fine-tuned with INT8 attention achieve higher accuracy on both GSM8k and MMLU benchmarks:
>
> **Table 7: Comparison on Qwen2.5-1.5B.**
>
> | All Using FP4 INference|Gsm8k $\uparrow$|Mmlu $\uparrow$ |
> | :---------------------:|:--------------:|:-------------: |
> | INT8 Fine-tuning | **0.5232** | **0.4934** |
> | FP8 Fine-tuning | 0.5031 | 0.4689 |
>
> **Table 8: Comparison on Qwen2.5-3B.**
>
> | All Using FP4 INference|Gsm8k $\uparrow$|Mmlu $\uparrow$ |
> | :---------------------:|:--------------:|:-------------: |
> | INT8 Fine-tuning | **0.5945** | **0.6032** |
> | FP8 Fine-tuning |0.5868| 0.5907 |
>
> These results indicate that INT8 attention not only offers broader hardware compatibility but also produces more accurate gradients during training, leading to better downstream performance when combined with FP4 inference.
>
>
> ---
>
> **If you feel your concerns have been resolved, we would greatly appreciate it if you consider raising the score. We will add all the experiments and analysis in the rebuttal to our paper.**

---

> > ### Comment · Reviewer_wcep · 2025-08-04
> >
> > Thank you for the detailed response and also for the new data. Most of my questions have been addressed, so I raised my rating.

---

> > > ### Author Response · Authors · 2025-08-05
> > > **Thank You for Your Positive Feedback**
> > >
> > > Thank you for your positive feedback and score update. Your insightful comments have greatly improved the quality of our work.

---

### Official Review · Reviewer_Czmu · 2025-07-02

**Clarity:** 3
**Significance:** 2
**Originality:** 3
**Rating:** 4
**Confidence:** 5

**Summary:**

The paper proposes a new low-precision attention model for LLM inference and training. A 4-bit floating point format (NVFP4) with a scaling factor is used for most of the attention parameters. To better match the dynamic range of all parameters when FP4 is used for the P parameter, two-level scaling is applied. Moreover, low-precision attention is demonstrated for the backward pass, using int8 integers for most matrix multiplications. This approach has been shown to be efficient for fine-tuning, but not suitable for training LLMs from scratch to design new pre-trained models. The new approach, combined with hardware optimizations such as fusing quantization with online softmax, results in a speedup over previous FlashAttention and Sliced Attention models.

**Questions:**

1- How is the value 480*6 selected for the range of P in first-level quantization?

2- To better understand the speedup compared to other models, it is suggested to present a trade-off plot between speed and performance degradation for various experiments.

**Ethical Concerns:**

["NO or VERY MINOR ethics concerns only"]

**Final Justification:**

The authors provide a comparison of the new approach with the previous FP4 design and also offer an analytical comparison with FlashAttention-3. In both cases, the new approach demonstrates better performance. As a result, I raised my score.

**Limitations:**

yes

**Paper Formatting Concerns:**

There are no paper formatting concerns

**Quality:**

3

**Strengths And Weaknesses:**

Strengths:

1. The paper is well-written and well-organized.
2. The int8 quantization for fine-tuning LLMs is interesting and novel.

Weaknesses:

1- The author is required to compare the new approach with previous work on FP4 and INT4 quantization of attention, such as [1, 2, 3, 4]. Comparisons for inference are not presented.
[1] Lin, Yujun, et al. "Qserve: W4a8kv4 quantization and system co-design for efficient llm serving." arXiv preprint arXiv:2405.04532 (2024).

[2] Liu, Lian, et al. "COMET: Towards Practical W4A4KV4 LLMs Serving." Proceedings of the 30th ACM International Conference on Architectural Support for Programming Languages and Operating Systems, Volume 2. 2025.

[3] Wang, Ruizhe, et al. "Optimizing Large Language Model Training Using FP4 Quantization." arXiv preprint arXiv:2501.17116 (2025).

[4] Frantar, Elias, et al. "Gptq: Accurate post-training quantization for generative pre-trained transformers." arXiv preprint arXiv:2210.17323 (2022).

2- Comparison with FlashAttention-3 is not presented. This comparison can be performed theoretically rather than by running on GPU.

3- The author is suggested to explain the reason behind the significant accuracy degradations in FScore for Hunyuan Video and Mochi when comparing Sege Attention3 (4-bit) vs. Sege Attention2 (8-bit).

---

> ### Author Rebuttal · Authors · 2025-07-30
>
> Dear Reviewer Czmu,
> Thank you for your valuable suggestions and questions. Below, we address each point raised.
>
> ---
> **W1.** The author is required to compare the new approach with previous work on FP4 and INT4 quantization of attention, such as [1, 2, 3, 4]. Comparisons for inference are not presented.
>
> **Response:** Strictly speaking, the methods mentioned above are not directly comparable, similar to how FP8-quantized FlashAttention3 is not compared against KV-cache-only quantization methods. To be specific, these methods do not quantize $Q$, $K$, and $V$ entirely to 4-bit, nor do they perform attention computations at 4-bit. Our work is the first to leverage hardware FP4 Tensor Core to perform the entire attention computation in FP4, accelerating the attention computation beyond just reducing memory usage. In contrast, existing methods still execute attention in FP16/BF16 and focus primarily on reducing the KV-cache memory, which can save GPU memory but does not improve performance for compute-bound workloads such as ViTs or LLM pre-filling.
>
> Nevertheless, to provide a more complete evaluation, we add two sets of comparison experiments. First, we compare SageAttention3 with QServe and Atom (please note that GPTQ is irrelevant here as it does not modify attention) on Qwen2.5-1B. As shown in the Table below, SageAttention3 achieves the best accuracy while also providing the fastest pre-filling speed. For decoding speed, because Atom and QServe cannot be compiled on the RTX5090, we report their theoretical decoding speedup.
>
> **Table 1: Comparison on Qwen2.5.**
>
> | Method |wikitext PPL $\downarrow$|LAMBDA $\uparrow$|Prefilling(64K) Latency $\downarrow$|Theoretical decoding speedup $\uparrow$ |
> | ---------|------------------------:|----------------:|-----------------------------------:|--------------------------------------: |
> | Qserve | 10.90296 | 0.631| 10.20257 | 4x |
> | Atom | 10.89669 | 0.631| 10.22090 | 4x |
> | SageAttn3| **9.56827** | **0.726** | **6.0989s** | 4x |
>
> Second, we compare SageAttention3 and QServe on CogVideoX-2B. As shown in the Table below, SageAttention3 not only achieves higher accuracy but also provides substantial speedup.
>
> **Table 2: Comparison on CogVideoX.**
>
> | Method |CLIPSIM ↑|VQA-a ↑|VQA-t ↑|FScore ↑|Latency $\downarrow$ |
> | ---------|--------:|------:|------:|-------:|-------------------: |
> | Qserve | 0.1683|19.7863|49.4831| 2.4488| 65s |
> | Atom | 0.1706|20.0896|46.9320| 2.4024| 64s |
> | SageAttn3| **0.1881** |**69.8600**|**70.3640**| **4.0350** | **27s** |
>
>
>
>
> ---
> **W2.** Comparison with FlashAttention-3 is not presented. This comparison can be performed theoretically rather than by running on GPU.
>
> **Response:** Thank you for suggesting a theoretical comparison with FlashAttention3. Indeed, a direct empirical comparison is not feasible because FlashAttention-3 is only supported on H100 GPUs. Instead, we provide a theoretical comparison based on the Tensor Core TOPS for different precisions as reported in NVIDIA’s official documentation. Using these hardware specifications, we estimated the theoretical compute speed (TOPS) of FlashAttention-3 and our SageAttention3 on GPUs that support FP4 Tensor Cores (B300, B200, and RTX 5090). The results are summarized below:
>
> **Table 3: Theoretical compute speed (TOPS) comparison.**
>
> |Method|B300 TOPS $\uparrow$|B200 TOPS $\uparrow$|RTX5090 TOPS $\uparrow$|
> |---|---|---|---|
> |FlashAttn3|2500|2500|209.5|
> |FlashAttn3 (FP8)|5000|5000|419|
> |SageAttn3 (FP4)|**15000**|**10000**|**1676**|
>
> These theoretical estimates show that SageAttention3 achieves significantly higher peak throughput on supported hardware, highlighting its potential for accelerating attention beyond FlashAttention-3.
>
>
>
>
> ---
> **W3.** The author is suggested to explain the reason behind the significant accuracy degradations in FScore for Hunyuan Video and Mochi when comparing SageAttention3 (4-bit) vs. SageAttention2 (8-bit).
>
> **Response:** Thank you for the valuable suggestion. The drop in FScore can be explained rather straightforwardly: FScore is highly sensitive to video quality degradation, and SageAttention3 (4-bit) inherently has lower numerical accuracy than SageAttention2 (8-bit). To better understand this effect, we performed an ablation study on the quantization precision of the $Q, K, P, V$ matrices and reported results across all layers of CogVideoX-2B, as shown in the table below. These results show that quantizing all $Q, K, P, V$ matrices to FP4 (SageAttention3) leads to noticeably higher errors compared to SageAttention2 (8-bit), which uses INT8 for $Q, K$ and FP8 for $P, V$. Lastly, we want to emphasize that although SageAttn3 slightly reduces video quality (visually negligible), it significantly boosts generation speed.
>
> **Table 4: Ablation of attention accuracy when Q K P V in different precisions.**
>
> | Method |Cossim $\uparrow$|L1 Error $\downarrow$|RMSE $\downarrow$ |
> | --------------------------------------------|-----------------|---------------------|----------------- |
> | $Q,K$ in FP4 |0.997848 |0.03883 |0.04104 |
> | $P,V$ in FP4 |0.998853 |0.04044 |0.03976 |
> | $Q,K$ in INT8 |0.999958 |0.00778 |0.00986 |
> | $P,V$ in INT8 |0.999953 |0.00813 |0.01510 |
> | $Q,K,P,V$ in FP4 (SageAttn3) |**0.99551** |**0.06923** |**0.07748** |
> | $Q,K$ in INT8; $P,V$ in FP8 (SageAttn2_8bit)|**0.99995** |**0.00849** |**0.00915** |
>
>
>
> ---
> **Q1.** How is the value 448*6 selected for the range of P in first-level quantization?
>
> **Response:** In standard FP4 quantization, the scale factor is typically defined as $S_{\bf P}=\frac{\mathrm{max}(|\bf P|)}{6}$, where 6 corresponds to the maximum representable value in E2M1 (FP4). However, since the values in $\bf P$ lie in the range $(0,1]$, the resulting the range of scale factor in $[0, 1/6]$, which is significantly smaller than the range of scale factor in E4M3 (FP8), which is $[-448, 448]$.
>
> To better utilize the FP8 range, we propose first to quantize $\bf P$ to a bigger range:
>
> $$P=\frac{\bf P}{S_{\bf P_1}}, ~~~ S_{P_1}=\frac{\mathrm{rowmax}(\bf P)}{448 \times 6}$$
>
> This ensures that the range of $\bf P$ is in $[0, 448 \times 6]$, enabling the second-stage FP4 quantization to use a scale factor of:
> $$S_{\bf P}=\frac{\mathrm{max}(\bf P)}{6} \in [0, 448]$$
>
> which better leverages the representational range of the E4M3 (FP8) format.
>
> In summary, 448×6 is based on the fact that the maximum representable values for E4M3 and NVFP4 are 448 and 6, respectively.
>
>
>
> ---
> **Q2.** To better understand the speedup compared to other models, it is suggested to present a trade-off plot between speed and performance degradation for various experiments.
>
> **Response:** Thank you for the suggestion. To illustrate the trade-off between accuracy and speed, we recorded the accuracy (CosSim) of various attention methods across all layers of CogVideoX-2B, along with their theoretical throughput on RTX 5090 and H100 GPUs. These results are summarized in the table below. We will update our manuscript to include this table and the corresponding trade-off plot.
>
> **Table 5: Speed and accuracy trade-off of different methods.**
>
> | Method |TOPS on 5090 $\uparrow$|TOPS on H100 $\uparrow$|Accuracy $\uparrow$ |
> | -----------------------|-----------------------|-----------------------|------------------- |
> | FlashAttention2 |214 |338 |100.000% |
> | FlashAttention3 (16bit)|N/A |470 |100.000% |
> | SageAttention1 |479 |518 |99.996% |
> | SageAttention2 (8bit) |643 |885 |99.995% |
> | FlashAttention3 (8bit)|N/A |890 |98.570% |
> | SageAttention3 (4bit) |1038 |N/A |99.551% |
>
>
>
> ---
>
> **If you feel your concerns have been resolved, we would greatly appreciate it if you consider raising the score. We will add the experiment and analysis in the rebuttal to our paper.**

---

> > ### Author Response · Authors · 2025-08-07
> > **Sincerely Looking Forward to Your Feedback**
> >
> > Thank you once again for your valuable comments. As the discussion period deadline approaches, we would greatly appreciate it if you could let us know whether our responses have adequately addressed your concerns.
> >
> > Best Regards,
> > The Authors

---

> ### Author Response · Authors · 2025-08-08
> **Looking Forward to Your Feedback (Under 21 Hours Left)**
>
> Dear Reviewer Czmu,
>
> ---
> Thank you for your important comments. With only 21 hours remaining in the discussion period, we would greatly appreciate your confirmation on whether our responses have adequately addressed your concerns.
>
> Best regards,
> The Authors

---

### Official Review · Reviewer_DSm8 · 2025-07-03

**Clarity:** 2
**Significance:** 3
**Originality:** 2
**Rating:** 4
**Confidence:** 2

**Summary:**

This paper studies doing forward and backward propagation for Transformer attention efficiently with low-bit precision. It proposes per-block 4-bit microscaling and a two-stage softmax quantization to compress both the query-key and attention-value matmuls. Experiments show that SageAttention3 runs 5× faster than FlashAttention while preserving end-to-end quality metrics across diverse models and SageBwd achieves lossless performance in fine-tuning
tasks, though full-scale pre-training still faces challenges.

**Questions:**

1. Accumulated quantization drift across layers? Small errors per layer can compound over dozens of layers, leading to distributional shift and degraded convergence. Some discussion on analysis or mitigation to control this drift will be appreciated.
2. The max-based scale factors seem vulnerable to outliers (sec 3.1). A single extreme outlier in a block dominates the scale and shrinks all other values toward zero. I was wondering if the authors considered any mechanism to mitigate the potential sensitivity to outliers?

**Ethical Concerns:**

["NO or VERY MINOR ethics concerns only"]

**Final Justification:**

Post-rebuttal final justification: Thank you for your in-depth rebuttal. I appreciate the discussion and analysis on masking, quantization design choices, error drift, and smoothing methods. I will maintain the original score.

**Limitations:**

yes

**Quality:**

3

**Strengths And Weaknesses:**

**Strengths**:

The proposed method introduces a principled two-level quantization and selective FP16 retention that balance accuracy and speed and leverages FP4 tensor cores with triton kernels. Validates both inference and training across diverse Transformer architectures and tasks, showing consistent efficiency improvements.

**Weaknesses**:

1. The optimization strategies proposed by this paper seem to be quite hardware specific, e.g., k-permutation tuned to NVIDIA Blackwell’s FP4 tensor cores. The optimizations may not map directly to other GPUs (e.g. AMD or older NVIDIA), or to TPUs.
2. The proposed pipeline assumes unmasked rows. Real Transformer use-cases often require causal masks (e.g., autoregressive language modeling). More discussions on how to integrate mask logic into the pipeline will be appreciated.
3. Heuristic INT8/FP16 backward mix: the proposed method only quantizes five of six backward matmuls to INT8, but keep the remaining one FP16. This choice is presented as empirical, with no systematic analysis of which matmuls are safe to quantize or why exactly dOV^T must remain in FP16. More discussions and insights into this will make the paper stronger.

---

> ### Author Rebuttal · Authors · 2025-07-30
>
> Dear Reviewer DSm8,
> Thank you for your valuable suggestions and questions. Below, we address each point raised.
>
> ---
> **W1.** The optimization strategies proposed by this paper seem to be quite hardware specific, e.g., k-permutation tuned to NVIDIA Blackwell’s FP4 tensor cores. The optimizations may not map directly to other GPUs (e.g. AMD or older NVIDIA), or to TPUs.
>
> **Response:** We'd like to clarify three aspects:
>
> 1. **Generality of Core Methods:**  The main techniques we propose, such as low-bit matrix quantization for faster computation, two-level quantization, removing explicit max operations for per-token quantization of the $P$ matrix, and outlier smoothing—are algorithmically general and can be adapted to different hardware platforms.
> 2. **Hardware-Specific Optimizations as a Necessity:**  Achieving peak FLOPS on cutting-edge systems inevitably requires some hardware-aware optimizations. For example, FlashAttention-3 (compared to FlashAttention-2) extensively exploits Hopper-specific features such as Tensor Core asynchrony and Tensor Memory Accelerators (TMA).
> 3. **Adaptable Principles:** While certain implementations (e.g., k-permutation) are architecture-dependent, the underlying principles (exploiting the tensor layout of low-bit MMA instructions) remain relevant across platforms, guiding adaptations for other hardware.
>
> ---
> **W2.** The proposed pipeline assumes unmasked rows. Real Transformer use-cases often require causal masks (e.g., autoregressive language modeling). More discussions on how to integrate mask logic into the pipeline will be appreciated.
>
> **Response:** SageAttention3 already supports efficient causal masking, as evidenced by the attention speeds reported in Figures **4-5** in the original paper. We are sorry that we omitted the implementation details in our paper (like FlashAttention). The solution is:
>
> 1. **Block Skipping**: Q/K blocks entirely masked by causality are excluded from memory loads.
> 2. **Partial Mask Handling**: For partially masked blocks loaded in the attention kernel, we apply $-\infty$ to the corresponding positions in the $P$ matrix to enforce causal masking.
>
> To validate effectiveness, we add an experiment that compares the end-to-end metrics and attention speed on Llama3.2-1B model, confirming that SageAttention3 maintains accuracy and enhances the efficiency in tasks with causal masks:
>
> **Table 1: FP4 inference result on an autoregressive language model.**
>
> |Method|wikitext PPL $\downarrow$|LAMBDA $\uparrow$|Prefilling(64K) Latency $\downarrow$|
> |-|-:|-:|-:|
> |BF16 Inference|9.29854|0.741|10.2145|
> |SageAttn3 Inference| 9.56827|0.726|6.09898|
>
> ---
> **W3.** Heuristic INT8/FP16 backward mix: the proposed method only quantizes five of six backward matmuls to INT8, but keep the remaining one FP16. This choice is presented as empirical, with no systematic analysis of which matmuls are safe to quantize or why exactly dOV^T must remain in FP16. More discussions and insights into this will make the paper stronger.
>
> **Response:** Thank you for the insightful question. We conduct an experiment and provide a theoretical analysis to explain why dOV^T remains in FP16.
>
> **Experiment**: The backward pass involves 5 MatMuls. ${\bf S}\_{ij} \gets {\bf Q}_i\mathbf {\bf K}_j^\top$, whose accuracy is fully addressed in SageAttention2. The remaining four are as follows:
>
> (1) ${\bf dP}\_{ij}\gets {\bf dO}_i {\bf V}_j^\top$
>
> (2) ${\bf dQ}_i\gets{\bf dQ}_i+{\bf dS}\_{ij}{\bf K}_j$
>
> (3) ${\bf dK}_j\gets{\bf dK}_j+{\bf dS}\_{ij}^\top {\bf Q}_i$
>
> (4) ${\bf dV}_j\gets{\bf dV}_j+{\bf P}\_{ij}^\top {\bf dO}_i$
>
> Among these, the last three matmuls directly add their results to the final output. However, **(1)** involves further computation:
> $$
> {\bf dS}\_{ij}\gets {\bf P}\_{ij}\circ ({\bf dP}\_{ij}-{\bf D}_i),
> $$
>
> where ${\bf dS}\_{ij}$ subsequently contributes to both ${\bf dQ}_i$ and ${\bf dK}_j$ in steps (2) and (3). Since the output of ${\bf dO}_i {\bf V}_j^\top$ is used in downstream matmuls, it is reasonable to keep this computation in higher precision (FP16).
>
> Empirical data support this conclusion. Using tensors from all layers of CogVideoX-2B, we evaluate the accuracy by quantizing each Matmul ("-" indicates the calculation in that row is not related to the Matmul in that column.):
>
> **Table 2: Attention gradient accuracy ablation.**
>
> |Matmul not quantized|(1)|(2)|(3)|(4)|None|
> |-|-|-|-|-|-|
> | $\bf dQ$'s Cossim $\uparrow$|**0.9202**|0.8152|-|-|0.8125 |
> | $\bf dQ$'s L1 Error $\downarrow$|**0.3308**|0.6326|-|-|0.6399 |
> | $\bf dQ$'s RMSE Error $\downarrow$|**5.0905**|8.5137|-|-|8.5827 |
> | $\bf dK$'s Cossim $\uparrow$|**0.9858**|-|0.9839|-|0.9839 |
> | $\bf dK$'s L1 Error $\downarrow$|**0.1826**|-|0.2003|-|0.2014 |
> | $\bf dK$'s RMSE Error $\downarrow$|**6.3992**|-|6.8168|-|6.8180 |
> | $\bf dV$'s Cossim $\uparrow$|-|-|-|**0.9982**|0.9981 |
> | $\bf dV$'s L1 Error $\downarrow$|-|-|-|**0.0901**|0.1220 |
> | $\bf dV$'s RMSE Error $\downarrow$|-|-|-|**1.1519**|1.2113 |
>
> We can see that keeping matmul (1) in FP16 leads to significant precision improvements.
>
> **Theoretical Analysis.** Following [1], assume that any matrix ${\bf X}\in\mathbb R^{n\times d}$ (e.g. ${\bf Q},{\bf K},{\bf V},{\bf dO}$) satisfies:
>
> - The entries in ${\bf X}$ are mutually independent.
> - ${\bf X}\_{ij} \sim N(\mu_{{\bf X},j},\sigma_{{\bf X},j}^2)$, i.e. the distribution of each token is identical.
>
> The quantization error of a matrix ${\bf X}$ is denoted as:
> $$
> \Delta{\bf X} := s_{\bf X}\hat{\bf X}-{\bf X},\quad \text{where } s_{\bf X},\hat{\bf X}=\psi({\bf X}).
> $$
>
> For example, consider the error in ${\bf dQ}$. Neglecting second-order error terms, we have:
> $$
> \Delta{\bf dQ}=\underbrace{({\bf P}\circ({\bf dO}\Delta{\bf V}^\top+\Delta {\bf dO}{\bf V}^\top)){\bf K}}_{\Delta {\bf dQ}^{(1)}\text{ from (1)}}+\underbrace{\Delta{\bf dS}{\bf K}+{\bf dS}\Delta{\bf K}}\_{\Delta{\bf dQ}^{(2)}\text{ from (2)}}
> $$
>
> Here, ${\bf dS}={\bf P} \circ ({\bf dP}-{\bf D})={\bf P} \circ ({\bf dO} {\bf V}^\top-{\bf D})$, where ${\bf D}={\bf dO}\odot {\bf O}$. In element-wise terms:
> $$
> {\bf dS}\_{ij}={\bf P}\_{ij} \sum_{k} {\bf dO}\_{ik} ({\bf V}\_{jk}-{\bf O}\_{ik})={\bf P}\_{ij} \sum_k {\bf dO}\_{ik} \left({\bf V}\_{jk}-\sum_l {\bf P}\_{il} {\bf V}\_{lk}\right).
> $$
>
> Since ${\bf V}$ is independent from other variables, by linearity of expectation:
> $$
> \mathbb E[{\bf dS}\_{ij}]=\mathbb E\left[{\bf P}\_{ij} \sum_k {\bf dO}\_{ik} \left(\mu_{{\bf V},k}-\sum_l {\bf P}\_{il} \mu_{{\bf V},k}\right)\right]=0.
> $$
>
> Moreover, as negating ${\bf V}$ flips the sign of ${\bf dS}\_{ij}$, the PDF of ${\bf dS}\_{ij}$ is symmetric. Using a "round-to-nearest" quantization policy, we have $\mathbb{E}[\Delta {\bf dS}]=0$. Thus
> $$
> \mathbb E\left[\Delta{\bf dQ}^{(2)}\right]=\mathbb E\left[\Delta {\bf dS} {\bf K}+{\bf dS} \Delta {\bf K}\right]=0,
> $$
>
> While $\mathbb{E}\left[\Delta {\bf dQ}^{(1)}\right]$ is generally non-zero (e.g., when distributions have non-zero means), indicating that ${\bf dQ}$'s error is dominated by $\Delta {\bf dQ}^{(1)}$.
>
> [1] AWQ: Activation-aware Weight Quantization for LLM Compression and Acceleration
>
> ---
> **Q1.** Accumulated quantization drift across layers? Small errors per layer can compound over dozens of layers, leading to distributional shift and degraded convergence. Some discussion on analysis or mitigation to control this drift will be appreciated.
>
> **Response:** Thank you for raising this insightful point. We agree that quantization errors can accumulate across layers. To investigate this,
>
> 1. We conduct an experiment using SageAttention3 on CogVideoX-2B and report the L1 error for each layer, as shown in the Table below. We observe that, in general, the accumulated error tends to increase with depth; however, it occasionally decreases in certain deeper layers, suggesting that some degree of error cancellation may occur.
> 2. To mitigate this drift, we try a simple approach: we set the three layers exhibiting the most significant error growth to use FP16 attention. As shown in the Table below, this simple strategy effectively reduces the overall error accumulation.
>
> **Table 3: Accumulated L1 error of layers.**
>
> ||Layer1 $\downarrow$|Layer10 $\downarrow$|Layer20 $\downarrow$|Layer30 $\downarrow$|
> |-|-|-|-|-|
> |Use SageAttn3 directly|0.0076|0.0922|0.1146|0.0571|
> |Keep 3 most sensitive layers to FP16|0.0076|**0.0447**|**0.0773**|**0.0429**|
>
> ---
> **Q2.** The max-based scale factors seem vulnerable to outliers (sec 3.1). A single extreme outlier in a block dominates the scale and shrinks all other values toward zero. I was wondering if the authors considered any mechanism to mitigate the potential sensitivity to outliers?
>
> **Response:** We agree with your observation; however, fine-grained control of every 1×16 outlier is challenging. Unlike quantization designed for model compression, our quantization is aimed at accelerating matrix multiplication, which requires performing the computation entirely in low precision while supporting direct dequantization. This constraint limits outlier control to either per-token or per-tensor granularity. Although approaches such as SmoothQuant and Hadamard transformations enable per-token or per-tensor control, we find them ineffective in our setting. SageAttention3 adopts the smoothing K and smoothing Q introduced in SageAttention2. To ablate these choices, we conduct ablation experiment on all layers of CogVideoX-2B, and the results are as below:
>
> **Table 4: Ablation of attention accuracy with different smooth methods.**
>
> |Method|Cossim $\uparrow$|L1 Error $\downarrow$|RMSE $\downarrow$|
> |-|-|-|-|
> |None|0.915642|0.33586|0.30348|
> |Smooth_Quant|0.930125|0.26761|0.25288|
> |Hadamard|0.941222|0.26207|0.22397|
> |Smoothing_Q|**0.982848**|**0.11565**|**0.12586**|
> |Smoothing_K|**0.991176**|**0.09483**|**0.09766**|
>
> ---
>
> **If you feel your concerns have been resolved, we would greatly appreciate it if you consider raising the score. We will add the experiment and analysis in the rebuttal to our paper.**

---

> ### Comment · Reviewer_DSm8 · 2025-08-05
>
> Dear authors,
>
> Thank you for your in-depth rebuttal. I appreciate the discussion and analysis on masking, quantization design choices, error drift, and smoothing methods. I will maintain the original score.
>
> Thanks,
>
> Reviewer DSm8

---

> ### Author Response · Authors · 2025-08-07
> **Thank you for you Feedback**
>
> We appreciate your valuable feedback and the time you’ve taken to review our work. We will incorporate the experiment and analysis from the rebuttal into the final manuscript.
> We hope our response has addressed your concerns and may help strengthen your evaluation score or confidence.
>
> Best Regards,
> The Authors

---

### Official Review · Reviewer_VcmM · 2025-07-04

**Clarity:** 2
**Significance:** 3
**Originality:** 3
**Rating:** 4
**Confidence:** 3

**Summary:**

The paper introduce SageAttention3, with two key contributions. First, they propose FP4 attention for inference and second, they design int8 backward pass for accelerated training. For FP4 inference, the authors propose microscaling combined with two-level scaling to better use FP8 scaling factors.

**Questions:**

1. How does the overall memory footprint of SageAttention3 compare to other existing works for long sequences?
2. Is there any shared implementation showing efficiency gains for both FP4 inference and int8 training together?

**Ethical Concerns:**

["NO or VERY MINOR ethics concerns only"]

**Final Justification:**

Considering authors' rebuttal and other reviewers' feedback, I increase my score.

**Limitations:**

Yes. The authors indirectly address their limitations in the future work section of their work.

**Quality:**

2

**Strengths And Weaknesses:**

Strengths:
- Exploiting low-precision FP4 acceleration is a compelling direction.
- Hardware-centric contribution including kernel-level optimizations
- Results prove the efficacy of the proposed method

Weaknesses:
- The two proposed contributions appears disconnected. While both broadly target low-precision methods, they employ different approaches. No experiment show the combined impact of the two methods. I think the INT8 training part is less mature and can be explored as a separate work later.
- There are limited experiments supporting efficiency claims for INT8 training.

---

> ### Author Rebuttal · Authors · 2025-07-30
>
> Dear Reviewer VcmM,
> Thank you for your valuable suggestions and questions. Below, we address each point raised.
>
> ---
> **W1.** The two proposed contributions appear disconnected. While both broadly target low-precision methods, they employ different approaches. No experiment show the combined impact of the two methods. I think the INT8 training part is less mature and can be explored as a separate work later.
>
> **Response:** Thank you for your insightful and important comment. Both contributions aim to improve attention efficiency through quantization, with FP4 inference focusing on accelerating inference and INT8 attention targeting training efficiency.
>
> To better demonstrate the connection between these two approaches, we add an experiment that combines them: first applying INT8 attention during fine-tuning, followed by FP4 attention during inference. Specifically, we fine-tuned Qwen2.5 for 1,000 steps using either BF16 or our INT8 attention, and then evaluated inference performance using our FP4 attention. The results on GSM8k and MMLU are shown below, where INT8 fine-tuning followed by FP4 inference yields higher accuracy, indicating that the two methods can be complementary, and their combination can deliver additional benefits. This improvement is likely because INT8 and FP4 share a more similar representable data distribution, reducing the mismatch error compared to BF16.
>
> **Table 1: Comparison on Qwen2.5-1.5B.**
>
> | All using FP4 Inference | Gsm8k $\uparrow$ | Mmlu $\uparrow$ |
> | :-----------------: | :--------------: | :-------------: |
> | BF16 Fine-tuning | 0.4912 |    0.4688    |
> | INT8 Fine-tuning | **0.5232** |  **0.4934**  |
>
> **Table 2: Comparison on Qwen2.5-3B.**
>
> | All using FP4 Inference | Gsm8k $\uparrow$ | Mmlu $\uparrow$ |
> | :---------------------: | :--------------: | :-------------: |
> |   BF16 Fine-tuning    |     0.5860     |    0.6000     |
> |   INT8 Fine-tuning    |   **0.5945**   |  **0.6032**   |
>
>
> ---
> **W2.** There are limited experiments supporting efficiency claims for INT8 training.
>
> **Response:** We would like to clarify that we have comprehensively evaluated the kernel efficiency and end-to-end efficiency in the original paper. First, we have assessed the attention kernel speed across different sequence lengths, head dimensions, and causal masking settings in the original paper. Specifically:
>
> - Forward pass speed of SageBwd is reported in Figures 15-16 (Appendix in the original paper).
> - Backward pass speed is detailed in Figures 17-18 (Appendix in the original paper).
> - Overall speed (forward + backward) is presented in Figures 6-7 (Main Text in the original paper).
>
> Second, we have evaluated and reported the end-to-end training efficiency gain of SageBwd in Table 4 in the original paper.
>
> Additionally, we appreciate the feedback regarding the need for more detailed efficiency analysis of INT8 training. To address this, we conducted additional experiments measuring the training latency (ms) breakdown of key components during training, using Llama3.1-8B and Qwen3-8B with sequence lengths of 16K and 32K. We find that:
>
> - SageAttn3 achieves about 30% speedup in attention computation compared to the original implementation.
> - End-to-end efficiency gains increase with sequence length, as attention latency becomes dominant with longer sequence lengths.
> - Total training latency is reduced by 13–15% across both models.
>
> **Table 3: Training latency (ms) breakdown on Llama3.1-8B and Qwen3-8B.**
>
> |Model (SequenceLength)|Method|FFN|Attention $\downarrow$|qkvo_projection|Others|Total $\downarrow$|
> |---|---|---|---|---|---|---|
> |Llama3.1-8B (16K)|FlashAttn2|105.97|388.30|388.88|2.03|885.18|
> |Llama3.1-8B (16K)|SageBwd|106.19|**269.47**|388.63|2.03|**766.3**|
> |Llama3.1-8B (32K)|FlashAttn2|214.01|1501.6|1504.82|4.08|3,224.5|
> |Llama3.1-8B (32K)|SageBwd|215.5|**1,036.9**|1498.09|4.09|**2,754.5**|
> |Qwen3-8B (16K)|FlashAttn2|93.37|391.07|389.82|2.03|876.29|
> |Qwen3-8B (16K)|SageBwd|93.51|**270.96**|389.02|2.02|**755.51**|
> |Qwen3-8B (32K)|FlashAttn2|184.13|1502.2|1501.8|4.08|3,192.3|
> |Qwen3-8B (32K)|SageBwd|184.9|**1037.5**|1497.7|4.08|**2,724.2**|
>
>
>
>
> ---
> **Q1.** How does the overall memory footprint of SageAttention3 compare to other existing works for long sequences?
>
> **Response:** The memory footprint of Sageattention3 is the same as existing comparable methods. Specially:
>
> (1) For compute-bounded tasks such as vision transformers or LLM prefilling, the comparable method is FlashAttention. The overall memory footprint of SageAttention3 is the size of the Q, K, and V of size $N\times d$ in FP16, saming with FlashAttention.
>
> (2) For memory-bounded LLM decoding, the overall memory footprint of SageAttention3 is the size of the Q of size $1\times d$ in FP4 and K, V of size $N\times d$ in FP4. Similar to other 4-bit KV cache quantization works, when applied to memory-bounded LLM decoding, SageAttention can quantize the KV cache to FP4, reducing memory usage and I/O bandwidth by 75% compared to standard FP16/BF16.
>
> In addition, we want to emphasize that **SageAttention3, similar to FlashAttention, is primarily designed for compute-bounded tasks such as vision transformers or LLM prefilling. In these scenarios, memory consumption remains minimal and does not impact attention latency.** Specifically:
>
> - **Memory usage:** SageAttention maintains the same memory footprint as FlashAttention, i.e., the size of the Q, K, and V matrices (**very small compared to the model parameters**). This is because the large intermediate matrices (e.g., the $N \times N$ attention scores $S = QK^\top$ and softmax matrix $P$) are never materialized in GPU memory, due to online softmax computation.
> - **Latency Impact:** The memory required for Q, K, and V is typically negligible compared to model parameters. Moreover, because attention is compute-bound, the I/O overhead for these matrices will be covered by computation and does not affect latency.
>
>
>
>
> ---
> **Q2.** Is there any shared implementation showing efficiency gains for both FP4 inference and int8 training together?
>
> **Response:** There are indeed shared implementation techniques that benefit both FP4 inference and INT8 training:
>
> 1. **Efficiency optimizations:** Both methods leverage (1) low-bit MMA PTX instructions to accelerate matrix multiplications, (2) eliminating the need for explicit max operations in the $P$ matrix by reusing both global and local maximum values from the online softmax computation (as introduced in Section 4.1), and (3) manipulating the memory layout of the two low-bit matrices involved in matrix multiplication to meet the requirements of the fastest MMA instruction.
> 2. Accuracy improvements: Both methods adopt key smoothing, where $K = K - \text{colmean}(K)$, to improve the numerical stability and precision of the attention computation.
>
> ---
>
> **If you feel your concerns have been resolved, we would greatly appreciate it if you consider raising the score. We will add the experiment and analysis in the rebuttal to our paper.**

---

> > ### Author Response · Authors · 2025-08-08
> > **Looking Forward to Your Feedback (Under 21 Hours Left)**
> >
> > Dear Reviewer VcmM,
> >
> > ---
> > Thank you for your important comments. With only 21 hours remaining in the discussion period, we would greatly appreciate your confirmation on whether our responses have adequately addressed your concerns.
> >
> > Best regards,
> > The Authors

---

> ### Author Response · Authors · 2025-08-07
> **Sincerely Looking Forward to Your Feedback**
>
> Thank you once again for your valuable comments. As the discussion period deadline approaches, we would greatly appreciate it if you could let us know whether our responses have adequately addressed your concerns.
>
> Best Regards,
> The Authors

---

### Author Response · Authors · 2025-08-04
**Looking Forward to Your Feedback**

Dear AC and Reviewers,

Thank you again for the valuable comments. We have carefully addressed the concerns in detail. We hope you find the response satisfactory. As the discussion phase is about to close, we look forward to any further feedback and are happy to clarify any remaining concerns.

---

### Note · Authors · 2025-08-15

Dear AC and Reviewers,

---
We sincerely thank all reviewers for their valuable feedback and have carefully addressed every concern:

- **Reviewer wcep and DSm8**: We greatly appreciate your positive constructive suggestions and feedback, which helped us improve the paper.
- **Reviewer VcmM and Czmu**: We appreciate your comments. Although we did not receive further textual feedback after our rebuttal, there were no additional questions, indicating that our clarifications effectively addressed your points.

We want to emphsize the **main contributions** of SageAttention3:

1. **First FP4 attention** for computation acceleration, achieving **1000+** TOPS on RTX5090.
2. **First trainable low-bit attention**, enabling accelerated training with lossless fine-tuning, and offering new insights into low-bit attention behavior during training.
3. **Unified training and inference acceleration**: Jointly using FP4 SageAttention and SageBwd not only accelerates both training and inference, but also delivers optimal efficiency and accuracy.

We thank the reviewers and AC again for their time and effort in evaluating our work.

---
Best regards,
The Authors

---

### Decision · Program_Chairs · 2025-09-17

**Decision:**

Accept (spotlight)

**Comment:**

This paper presents strong contributions in efficient FP4 inference and INT8 training. While reviewers initially raised significant concerns about the two parts feeling disconnected and lacking crucial comparisons, the authors delivered an exemplary rebuttal. They provided new experimental data that directly resolved all core issues, which successfully convinced three of the four reviewers to raise their scores. Due to the paper's technical merit and the authors' effective engagement in the discussion, the final recommendation is Accept.